# Optimal cancer evasion in a dynamic immune microenvironment generates diverse post-escape tumor antigenicity profiles

Jason T George[1,2,3]*, Herbert Levine[3,4,5]*

[1]Department of Biomedical Engineering, Texas A&M University, Houston, United States; [2]Engineering Medicine Program, Texas A&M University, Houston, United States; [3]Center for Theoretical Biological Physics, Rice University, Houston, United States; [4]Department of Physics, Northeastern University, Boston, United States; [5]Department of Bioengineering, Northeastern University, Boston, United States

**Abstract** The failure of cancer treatments, including immunotherapy, continues to be a major obstacle in preventing durable remission. This failure often results from tumor evolution, both genotypic and phenotypic, away from sensitive cell states. Here, we propose a mathematical framework for studying the dynamics of adaptive immune evasion that tracks the number of tumor-associated antigens available for immune targeting. We solve for the unique optimal cancer evasion strategy using stochastic dynamic programming and demonstrate that this policy results in increased cancer evasion rates compared to a passive, fixed strategy. Our foundational model relates the likelihood and temporal dynamics of cancer evasion to features of the immune microenvironment, where tumor immunogenicity reflects a balance between cancer adaptation and host recognition. In contrast with a passive strategy, optimally adaptive evaders navigating varying selective environments result in substantially heterogeneous post-escape tumor antigenicity, giving rise to immunogenically hot and cold tumors.

*For correspondence:
jason.george@tamu.edu (JTG);
h.levine@northeastern.edu (HL)

**Competing interest:** The authors declare that no competing interests exist.

## Editor's evaluation

This study presents a valuable mathematical model for the adaptive dynamics of cancer evolution in response to immune recognition. The mathematical analysis is rigorous and convincing, and overall the framework presented could be used in the future as a solid base for analytically tracking tumor evasion strategies. The work will be of interest to evolutionary cancer biologists and potentially may also have implications for the design of clinical interventions.

## Introduction

Cancer dynamics, encompassing both genotypic evolution and phenotypic progression, lies at the heart of treatment failure and disease recurrence, and therefore represents a significant and stubborn therapeutic hurdle. Prior research efforts have made substantial progress in detailing the mathematics of acquired drug resistance (*Iwasa et al., 2006*; *Michor et al., 2004*; *Komarova, 2006*) and the complementary roles of phenotypic and genotypic changes (*Gupta et al., 2019*). Recently, there has been much renewed interest in therapies that utilize the adaptive immune system to confer durable remission (*Couzin-Frankel, 2013*; *Waldman et al., 2020*). These latter breakthroughs have generated considerable interest in quantifying the cancer-immune interaction (*Mayer et al., 2019*; *Sontag,*

2017; *George et al., 2017*). As with targeted therapeutic resistance via compensatory evolution or adaptive rewiring (*Bergholz and Zhao, 2021*), tumors can similarly evade the immune system via either elimination or downregulation of tumor-associated antigens (TAAs) normally detectable by the T cell repertoire (*Rosenthal et al., 2019*). However, several key features distinguish immune-specific evasion from classical drug resistance (*Komarova, 2006*). Dynamical changes in cancer genotypes and phenotypes, while problematic for conventional therapies, create additional TAAs that may subsequently be recognized by distinct T cells (*Yarchoan et al., 2017*). Thus, the evolving diversity of the T cell repertoire, consisting of billions of unique clones each with a distinct T cell receptor, provides adaptive immunity and immunotherapy the unique advantage of repeated tumor recognition opportunities (*George and Levine, 2021*; *Lakatos et al., 2020*; *Qi et al., 2014*), making long-term evasion more challenging.

Previous research efforts have investigated the diversity of evolutionary trajectories and the extent of cancer-immune co-evolution occurring in early disease progression (*George and Levine, 2018*; *George and Levine, 2020*). These works were based on increasing evidence of significant and sustained tumor evolution driven by immune surveillance (*Turajlic et al., 2018*; *Jamal-Hanjani et al., 2017*). Immunosurveillance via distinct T cell clones imposes an adaptive, stochastic recognition environment on developing cancer populations (*Desponds et al., 2016*) that can result either in cancer elimination, escape, or equilibrium (*Schreiber et al., 2002*; *Dunn et al., 2004*). Equilibrium results in cancer co-existence with the immune system over large time scales (*Turajlic et al., 2018*), thereby motivating the need for a more complete understanding of the interplay between immune recognition and cancer evolution for effective therapeutic design. In addition to parsing this complexity, the precise extent to which a cancer population may *actively* evade repeated immune recognition attempts is at present unknown.

Previous modeling efforts have assumed that cancer adaptation occurs passively, that is, without behavior predicated on knowledge of the current immune microenvironment (IME). However, it is well known that cancer populations commonly undergo phenotypic changes capable of altering their immunogenicity (*Tripathi et al., 2016*); these changes could be coupled to sensing of the IME in a manner similar to cancer mechanical, chemical, and stress sensing (*Lee et al., 2019*; *Damaghi et al., 2013*; *Rosenberg, 2001*). Moreover, direct experimental evidence demonstrates genetic adaptation in bacterial systems capable of sensing stress and consequently varying the per-cell mutation rate (*Al Mamun et al., 2012*; *Rosenberg and Queitsch, 2014*); there appear to be similar stress pathways in cancer (*Bindra et al., 2007*). Therefore, an alternative to passive evolution is for cancer populations to actively sense and evade recognition in the current environment en route to metastasis in a manner that maximally benefits survival, which we refer to henceforth as the 'optimal escape hypothesis.' Understanding the extent and associated features of optimized tumor evasion is a crucial first step to identifying the best therapeutic approach, particularly for T cell immunotherapies that may be temporally varied.

Here, we introduce a mathematical framework, which we call 'Tumor Evasion via adaptive Antigen Loss' (TEAL), to quantify the aggressiveness of an evolutionary strategy executed by a cancer population faced with a varying recognition environment. This framework enables a dynamical analysis of both passive and optimized evasion strategies. The TEAL model describes a discrete-time stochastic process tracking the number of targets available to a recognizing adaptive immune system. We apply dynamic programming (*Bellman and Dreyfus, 1959*; *Ross, 2014*) in order to solve the corresponding time homogeneous Bellman equation detailing the tumor optimal evasion strategy for a specific example of the assumed penalty for attempting to avoid immune detection. In doing so, we obtain an exact analytical characterization of the evasion policy that maximizes long-run population survival, which we show is the unique solution. We can then quantify the enhancement in survival for optimal threats relative to their passive counterparts under a variety of temporally varying recognition environments. Surprisingly, we find that optimized strategies exhibit substantial diversity in their dynamical behavior, distinguishing them from threats with a fixed evolutionary strategy. Notably, immune recognition efficiency and the IME microenvironment are predicted to influence the likelihood for tumors to either accumulate or lose therapeutically actionable TAAs prior to their escape. The TEAL model represents a first attempt to explicitly represent – and in the future test – the optimal escape hypothesis in order to frame cancer evasion as a dynamic and informed strategy aimed at maximizing population survival.

## Model development

In greatest generality, our model consists of an evading clonal population that may be targeted over time by a recognizing system. We assume henceforth that the recognition-evasion pair consists of the T cell repertoire of the adaptive immune system and a cancer cell population, recognizable by a minimal collection of $s_n$ TAAs present on the surface of cancer cells in sufficient abundance for recognition to occur over some time interval $n$. Our focus is on a clonal population, recognizing that subclonal TAA distributions in this model may be studied by considering independent processes in parallel for each clone.

Experimental evidence and prior modeling suggest that tumors may be kept in an 'equilibrium' state of small population size prior to either escape or elimination, with repeated epochs of recognition and evasion (*Dunn et al., 2004*; *Turajlic et al., 2018*; *George and Levine, 2020*). We adopt a coarse-grained strategy and assume that during each epoch, the immune system has an opportunity to independently recognize each of the $s_n$ TAAs with probability $q$, and also the cancer population can lose recognized TAAs, each with probability $\pi_n$, which we refer to as the *antigen loss rate*. The antigen loss rate is either fixed or chosen by the cancer population using information available in the current period. If the immune system cannot detect any of the available TAAs in a given period, then the cancer population escapes detection. On the other hand, if $r_n > 0$ antigens are detected by the adaptive immune system in this time frame, then the cancer population is effectively targeted. This leads to cancer elimination unless the population is able to lose each of the $r_n$ recognized antigens during the same period. This loss of recognition would presumably arise in a subpopulation that would then expand at the expense of the successfully targeted cells. If evasion balances recognition and all detected antigens are lost, then equilibrium (non-escape, non-elimination) ensues, and the process repeats in the next period with a new number of target antigens given by a state transition equation

$$s_{n+1} = s_n - r_n + \beta + f_n \tag{1}$$

where $\beta$ represents the basal rate of new antigen accumulation, and $f_n$ represents the addition of new TAA targets dependent on the rate of escape $\pi_n$ in the current state. We shall refer to $f_n$ as the (intertemporal) penalty term, the idea being that changes that lead to antigen loss will out of necessity give rise to the creation of new TAAs, in the form of either overexpressed/mislocalized self-peptides or tumor neo-antigens.

The model therefore defines a discrete time process that involves changes to both the tumor and the immune system. The process ends in cancer elimination if the cancer population is unable to match all of the $r_n$ recognized antigens at any period. The process ends in cancer escape if at any period the number of recognized antigens is zero ($r_n = 0$). This framework mirrors the outcomes resulting from known tumor-immune interactions, a process that leads via immunoediting to cancer escape, elimination, or equilibrium (*Schreiber et al., 2002*; *Dunn et al., 2002*; *Dunn et al., 2004*; *Koebel et al., 2007*). Here, tumor antigenicity is represented by the total number of post-escape TAAs. We do not distinguish between different types of TAA loss, which may occur through a number of mechanisms, including somatic mutation, epigenetic regulation, or phenotypic alteration.

## Passive evader

In the passive case, the cancer population does not change its evasion rate so that $\pi_n = p$ is fixed and independent of any of the parameters governing the recognition landscape. For this case, we shall also use the simple assumption that the net antigen accumulation and penalty $\beta + f$ is a fixed constant.

## Optimal evader

In the optimized case, $\pi_n$ is chosen in order to maximize the overall evasion probability as a function of parameters realizable to the cancer at period $n$. We assume that $s_n$ the number of TAAs as well as $r_n$ the size of the recognized subset is knowable by the cancer prior to strategy selection. In addition, we postulate that the intertemporal penalty scales directly with $\pi_n$, a reasonable assumption given, for example, the direct relationship between mutagenesis and passenger mutation accumulation (*Pon and Marra, 2015*; *McFarland et al., 2014*). While many functional forms of $f_n(\pi_n, r_n, s_n)$ would be reasonable, we assume in general that the penalty is $\pi_n$-linear:

$$f_n(s_n, r_n, \pi_n) = h_m(s_n, r_n)\pi_n. \tag{2}$$

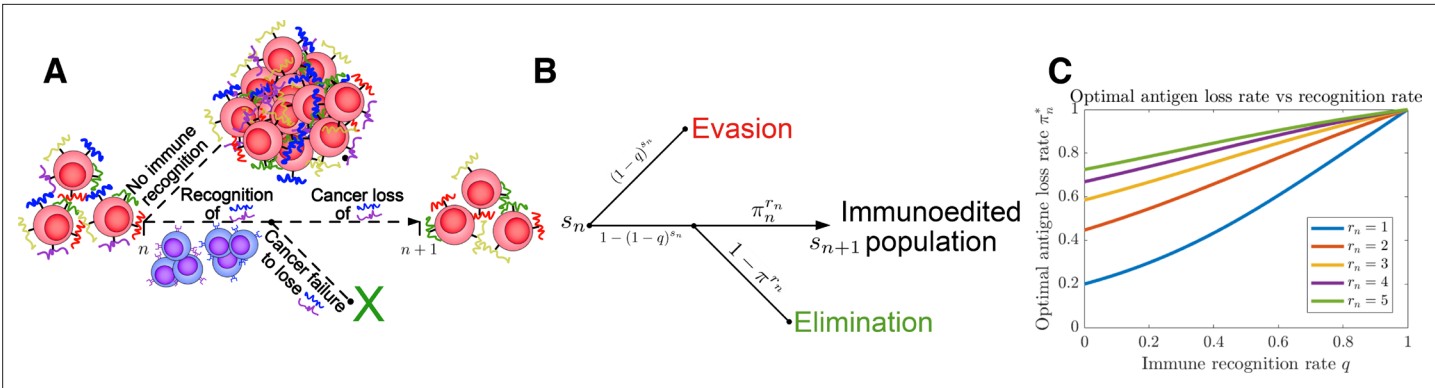

**Figure 1.** Tumor Evasion via adaptive Antigen Loss (TEAL) model. (**A**) Illustration of tumor antigen detection and downregulation in the TEAL model of cancer-immune interaction. (**B**) The directed graph with nodes representing the states of the TEAL model and edges labeled based on the probability of their occurrence. The interaction leads to elimination, equilibrium, or escape. Both evasion and elimination are absorbing states, and the equilibrium state results in repeated interaction. (**C**) Plots of single-period cancer optimal antigen loss rates $\pi^*$ given by **Equation 8** are plotted as a function of recognition rate $q$ for various numbers of recognized antigens $0 < r_n \le s_n$ with $s_n = 5$.

The online version of this article includes the following source data and figure supplement(s) for figure 1:

**Figure supplement 1.** Illustration of the Tumor Evasion via adaptive Antigen Loss (TEAL) model.

**Figure supplement 2.** Passive Evader outcome tree.

**Figure supplement 3.** Active Evader decision tree.

**Figure supplement 4.** Active Evader reward structure.

**Figure supplement 5.** Break-even evasion probability as a function of recognition probability.

**Figure supplement 5—source data 1.** Source data contains a spreadsheet of *Figure 1—figure supplement 5B* table.

**Figure supplement 6.** Probability of equilibrium.

**Figure supplement 7.** Mean antigenic load at equilibrium.

**Figure supplement 8.** Error of upper estimate $\tilde{\mu}$.

**Figure supplement 9.** Simulated and analytical mean antigenic load at equilibrium.

**Figure supplement 10.** Probability of Recognizer success.

**Figure supplement 11.** Optimal evasion.

To make our system analytically solvable, we use a specific choice in which $h_m$ scales monotonically as a function of both $r_n$ and $s_n$ and $h_m \propto r_n$ in the large $r_n$ limit (see 'Methods'). Since the number of recognizable (and thus actively targeted) TAAs reflect, all else being equal, an active IME hostile to cancer, we assume that subsequent total TAA addition, $\beta + f_n$, are dependent on the current level of immune detection, thereby taking into account the increased cost of surviving in, for example, an inflammatory IME. The temporal dynamics of the TEAL process are illustrated in *Figure 1A* and *Figure 1—figure supplement 1*.

## Varying environments

Using the above framework, we subject both passive and active cancer evasion tactics to temporally varying recognition profiles. We partition pre-escape dynamics into four cases based on immune recognition $q$ and basal TAA arrival $\beta$, from which we characterize the distribution of escape time, cumulative mutational burden, and predicted post-escape tumor immunogenicity.

## Results

The following section presents the main findings of our analysis (full mathematical details are provided in the 'Methods' section). For $s_n$ available and $r_n$ recognized TAAs, we have that $r_n \sim \text{Binom}(s_n, q)$. Conditional on recognition ($r_n > 0$), the number of downregulated antigens, $\ell_n$, is given by $\ell_n \sim \text{Binom}(r_n, \pi_n)$. Recognition therefore occurs with probability $\mathbb{P}(r_n > 0) = 1 - (1 - q)^{s_n}$. Similarly, non-elimination occurs following recognition with probability $\mathbb{P}(\ell_n = r_n) = \pi_n^{r_n}$. A decision

tree for the TEAL process is illustrated in *Figure 1B* (passive and active decision trees used in the analysis are depicted in *Figure 1—figure supplements 2–4*).

## Passive evasion strategy

For a passive evader, the TAA loss rate is fixed so that $\pi_n = p$. It can be shown (see Methods Section. Distribution of lost antigens) that the dynamics governed by *Equation 1* in the passive case can be represented by their mean trajectories while the cancer population is in equilibrium, given by

$$\mathbb{E}_n\left[S_{n+1}\right] = S_n - \frac{p(1-\gamma)\eta^{s_n-1}}{\eta^{s_n}-\gamma^{s_n}}s_n + (\beta + f), \tag{3}$$

where $\eta \equiv 1 - q(1-p)$ is the probability of equilibrium (non-escape, non-elimination) between the cancer and immune compartments for a single TAA given the existence of at least one available TAA. These dynamics may be approximated by

$$\mathbb{E}_n\left[s_{n+1}\right] \approx (1-q)s_n + (\beta + f), \tag{4}$$

where $\mathbb{E}_n\left[\cdot\right]$ is the conditional expectation given the information available at time $n$. The approximation given by *Equation 4* is a lower estimate of tumor antigenicity and is accurate as long as $p$ and $q$ are not both small and in particular for choices that give rise to large tie probability (*Figure 1—figure supplements 6 and 10*).

## Optimal evasion strategy

In contrast to the above case where $\pi_n$ was fixed at $p$, Here, the antigen loss rate is variable and selected optimally given the current state of total $s_n$ and recognized $r_n$ antigens. The use of dynamic programming to address the optimal long-term evasion policy relies on a defined value function (*Bellman and Dreyfus, 1959*). We shall focus on the case where the cancer population is assigned normalized values of 1 at any period resulting in escape and 0 otherwise. The corresponding stationary Bellman equation takes the form

$$J_n = \max_{\pi_n} \mathbb{E}_n\left[\pi_n^{r_n}\left[(1-q)^{s_{n+1}} + \left(1 - (1-q)^{s_{n+1}}\right)J_{n+1}\right]\right], \tag{5}$$

where the value function $J_n = J(s_n, r_n, \pi_n)$ represents the maximal attainable value at period $n$; (Methods Section Dynamic programming solution). It can be shown that

$$J_n = \frac{A_n\gamma^{s_n}}{1 - (1-q)^{s_n}} \tag{6}$$

with

$$A_n = \frac{\delta_n q(1-q)^{\beta+r_n/c-r_n}}{1 - \delta_n q(1-q)^{\beta+r_n/c-r_n}} \tag{7}$$

satisfies *Equation 5*. Here, $0 < \delta_n \leq 1$ is a free parameter that varies inversely with the risk aversion of the evader (larger values imply a bolder strategy). One advantage of the dynamic programming approach is that it reduces an infinite-period optimization problem to a sequence of single-period optimizations. The corresponding optimal policy is given by the sequence

$$\pi_n^* = \left(\frac{\delta_n q}{1 - (1-q)^{s_n}}\right)^{1/r_n}. \tag{8}$$

Plots of $\pi_n^*$ are given for various $r_n$ in *Figure 1C* and *Figure 1—figure supplement 11*. As expected, this closed-form strategy results in increased values for the optimal antigen loss rate $\pi_n^*$, which increase for increasing $q$ and $r_n$. We take $\delta_n = 1$ in subsequent analysis (so that the optimal strategy when $s_n = r_n = 1$ is $\pi_n^* = 1$).

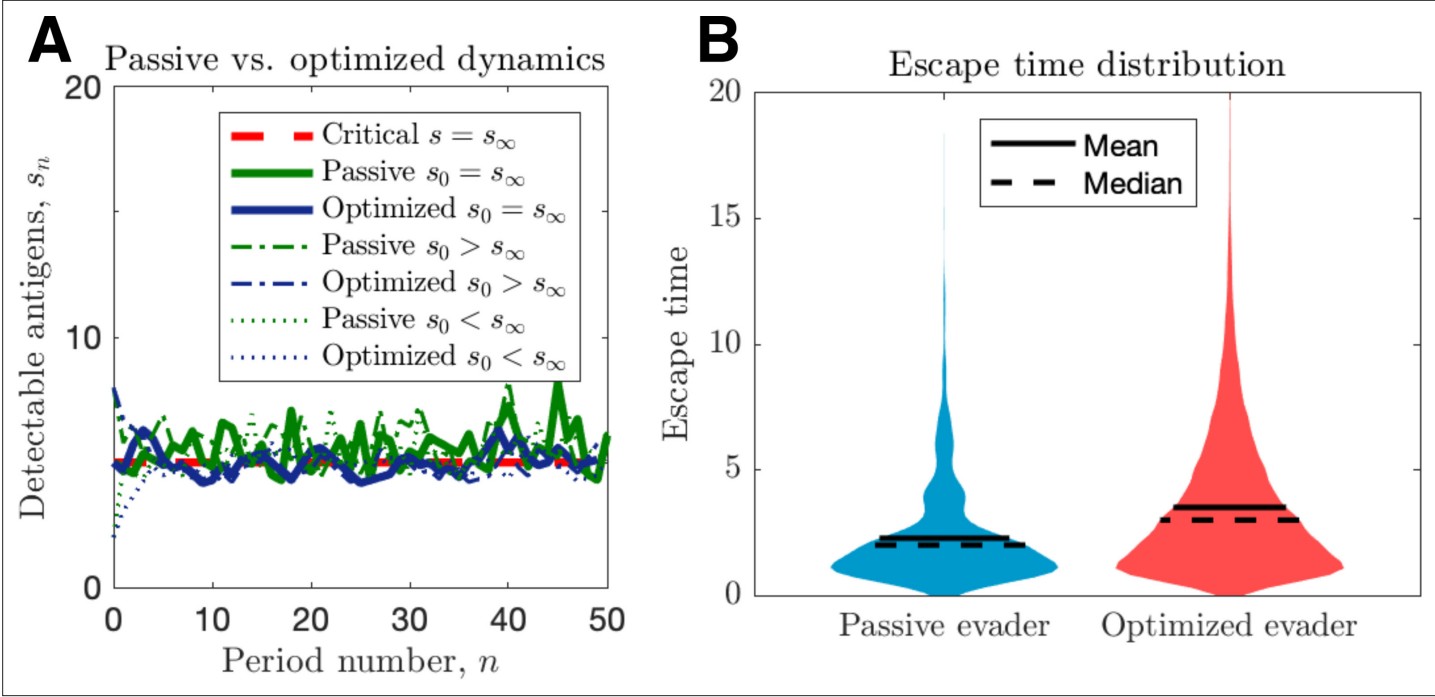

**Figure 2.** Passive and optimized evasion strategies against stationary threats. (**A**) Comparisons of the temporal dynamics of passive (green) and active (blue) strategies with parameter selections giving equal mean behavior. In the active case, $q$ yields stable dynamics, giving mean antigen arrival $\beta$. In the passive case, $p$ was selected to match the mean optimal evasion rate and the expected $s_n$ of the active case. Also, $f$ and $\beta$ both chosen so that $s_\infty$, and the results plotted for $s_0 \in \{2, 5, 8\}$. (**B**) $10^6$ replicates of this process were used to calculate distributions of stopping times conditioned on escape. This distribution generates passive (resp. optimized) $p_{\text{escape}}$ of 5.37 (resp. 8.44).

The online version of this article includes the following source data for figure 2:

**Source data 1.** Source data contains a spreadsheet of data for *Figure 2B*.

## Active evasion strategies enhance population survival rates

For a fixed TAA arrival, *Equations 3 and 4* describe a mean-reverting process. Consequently, the mean number of TAAs approaches a stable equilibrium

$$\lim_{n\to\infty} \mathbb{E}_n\left[s_{n+1}\right] \approx (\beta + f)/q. \tag{9}$$

as long as the cancer neither escapes nor is eliminated. In the optimal case, a similar equilibrium value $s_\infty$ may be calculated:

$$s_\infty = \frac{\beta}{q\left|1/\ln(1-q) - 1\right|}. \tag{10}$$

In this case, stability is more complex: If immune recognition is sufficiently effective, meaning $q > q^* = 1 - e^{-1}$, then *Equation 10* is a stable equilibrium exhibiting mean reversion similar to that of the passive case. On the other hand, recognition impairment ($q < q^*$) gives rise to an instability, which results in a system harboring an initial number of targets $s_0$ being driven either to escape if $s_0 < s_\infty$ or to large accumulations (and likely elimination) if $s_0 > s_\infty$ (*Figure 5—figure supplement 2*).

We proceed by contrasting active and passive escape rates assuming no recognition impairment, and discuss the implications of immune impairment in a later section. Simulations of passive and optimized strategies with passive evasion rates matching mean optimal evasion rates ($p = \mathbb{E}\left[\pi_n^*\right] |s_\infty$) are compared in *Figure 2*. Despite identical mean TAA evolution (*Figure 2A*) and comparable intertemporal penalties, the optimized strategy results in substantially higher cancer escape probability (150%) compared to the passive case. Moreover, optimized strategies generate wider escape time distributions, thus illustrating an adaptive evader's sustained effort to thwart elimination prior to escape (*Figure 2B*).

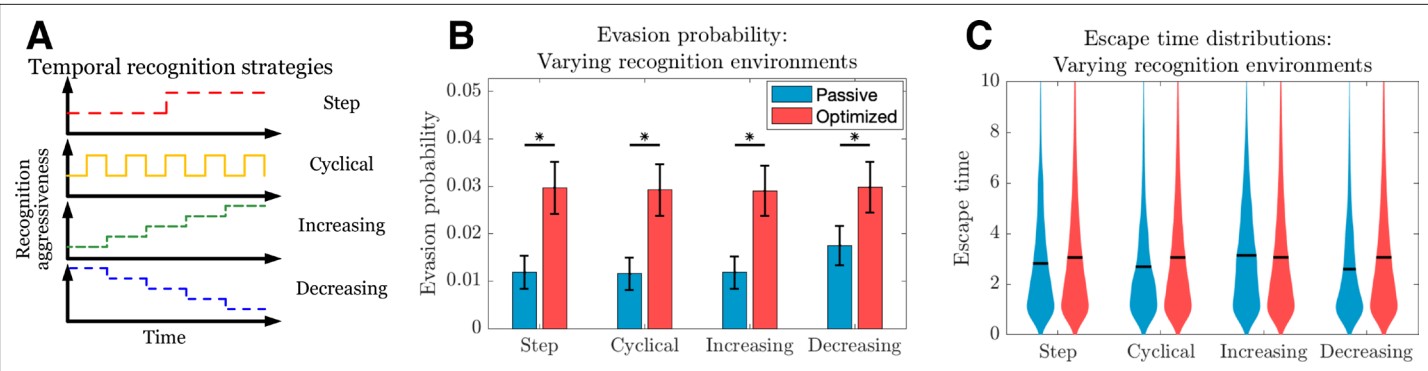

**Figure 3.** Passive and optimized evasion strategies for temporally varying recognition profiles. (**A**) Temporally varying recognition functions are selected and applied to threats employing passive (blue) and optimized (red) evasion strategies. (**B**) The mean and standard deviation of escape probabilities is compared across recognition profiles for each strategy (pairwise significance was assessed using two-sample *t*-test at significance $\alpha$ with p<10⁻⁵). (**C**) Escape time distributions are generated for step, cyclical, increasing, and decreasing recognition environments (solid line: mean). In each case, mean total new antigen arrival $\beta + \mathbb{E}[f_n]$ for passive (resp. optimized) evasion were 4.39 (resp. 4.75), and $10^3$ simulations of $10^3$ replicates each were used for statistical comparison; all samples were aggregated for escape time violin plots (solid line denotes mean).

The online version of this article includes the following source data for figure 3:

**Source data 1.** Source data contains a spreadsheet of data for *Figure 3B, C*.

## Arbitrary recognition landscape

The above describes the dynamics of passive and optimized cancer co-evolution during adaptive immune recognition with constant governing parameters. We can more generally apply this approach to understand how an evasion strategy affects the likelihood and timing of cancer escape under a variety of temporally varying recognition landscapes. Such landscapes could, for example, be imposed by a clinician temporally modulating an immunotherapeutic intervention and are routinely proposed in the setting of traditional therapies, where attempted strategies have included a variety of cyclical burst approaches (*Foo and Michor, 2009*; *Eigl et al., 2005*). A similar approach could be taken with regard to timing and dosage of adoptive T cell immunotherapy. An advantage of our dynamic programming approach is the ability to study optimal evasion strategies for arbitrary recognition landscapes (*Figure 3A*). We simulate TEAL dynamics and find that optimized immune evaders are more successful in evading detection than their passive counterparts across various recognition landscapes (*Figure 3B*). Evasion, when it occurs in the optimized case, does so largely after a sustained interaction with the recognizing threat (*Figure 3C*). Collectively, our results detail the dynamics of sustained cancer-immune co-evolution via TAA loss in threats capable of adopting adaptive evasion strategies in the presence of complex treatment modulation (*George and Levine, 2020*; *Turajlic et al., 2018*).

## Optimal evaders under effective immune recognition accrue mutations at a fixed rate

One consequence of mean reversion is that the rate of mutation accumulation over time, $\lambda(n)$, is linear in $n$ (Methods Section Mean optimal transitions):

$$\lambda(n) = \frac{2\beta \ln(1-q)}{1 - \ln(1-q)}n, \quad q > q^* = 1 - e^{-1}. \tag{11}$$

The prediction of constant accumulation is consistent with empirically observed cancer mutation behavior (*Lawrence et al., 2013*; *Alexandrov et al., 2013*). This is not what holds in the impaired case (as will be discussed later), thus suggesting that early cancer progression often proceeds in an environment with effective immune recognition. Additionally, our formula shows that larger mutation rates can be caused by large evasion penalties or by reduced immune recognition. Of course, the TEAL model does not consider any specific features that determine the values of the effective parameters. Instead, its utility is in quantifying the overall effect of reducing antigen detection resulting from, for example, transitions to an immune impaired microenvironment.

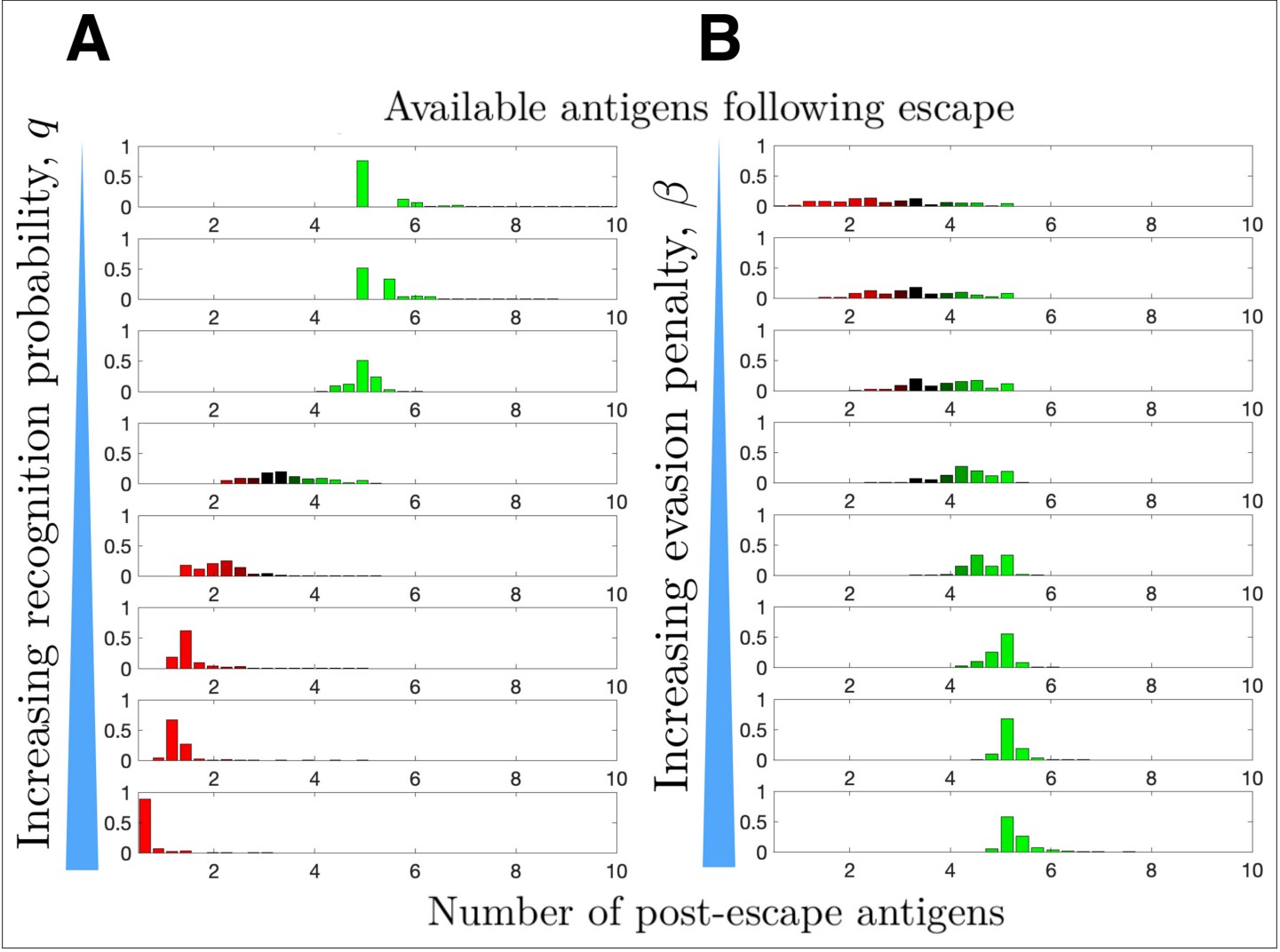

**Figure 4.** Distribution of available post-escape tumor antigens. The distribution of tumor-associated antigens (TAAs) was estimated from simulations of optimized cancer evasion resulting in escape and plotted for (**A**) increasing recognition probability $q \in \{0.6, 0.65, 0.7, 0.75, 0.8, 0.85, 0.9, 0.95\}$ and (**B**) increasing evasion penalty $\beta \in \{0.1, 0.2, 0.3, 0.4, 0.5, 0.6, 0.7, 0.8\}$. For (**A**), $\beta = 0.59$. For (**B**), $q = 0.7 > q^*$. In both cases, $s_\infty = 5$ and $n = 10^6$ simulations were performed for each histogram.

The online version of this article includes the following source data for figure 4:

**Source data 1.** Source data contains a spreadsheet of data for *Figure 4A, B*.

### Post-escape tumor antigenicity determined by a balance between recognition aggressiveness and local penalties in the immune microenvironment

The prior section related recognition and penalty to observed mutation rates. We now consider their combined effects on tumor immunogenicity following immune escape. The TEAL model represents immunogenicity by the number of available TAAs at the time of cancer detection, an important predictor of immunotherapeutic efficacy (*Martin et al., 2016*; *Samstein et al., 2019*; *Goodman et al., 2017*). We apply the TEAL model to simulate evading cancer populations, focusing exclusively on trajectories that result in tumor escape, to characterize the distribution of available TAAs. This is performed first for increasing immune recognition rates $q$ (*Figure 4A*) and then for increasing penalty term $\beta$ (*Figure 4B*). Our results demonstrate that larger penalties result in higher post-escape TAA levels, while efficient immune recognition depletes available TAAs. The presumptive reason for this latter observation is that escape in the presence of strong immune recognition biases the tumor to have low numbers of TAAs. This prediction agrees with recent empirical observations that strong

immune selective pressure in early cancer development results in tumor neoantigen depletion and is prognostic of poor clinical outcome (*Rosenthal et al., 2019*; *Lakatos et al., 2020*).

## Variation in the tumor microenvironment drives the generation of immune hot vs. cold tumors under optimal evasion

In the passive evader case, antigenicity fluctuates around a stable equilibrium that varies directly with penalty and inversely with recognition. The adaptive case gives rise to more complex behavior resulting from impairments in immune recognition or changes in penalty (*Figure 5—figure supplements 1 and 2*). These changes are important manifestations of disease progression, which may alter the immunogenic landscape via impairments in immune recognition, such as MHC downregulation, co-stimulation alteration, T cell exclusion, or the establishment of a pro-tumor IME, via. for example. M2 macrophage polarization (*Liu et al., 2021*; *Goswami et al., 2017*). Although many factors may affect recognition rates, for simplicity we shall refer to larger vs. smaller immune recognition rates $q$ as *infiltrated* vs. *excluded*.

On the other hand, the generation of new TAA targets is expected to vary substantially across tumor type, for example, due to differing somatic mutation rates. Within a given tumor subtype, variations in the hostility of the IME, resulting from a large variety of possible mechanisms (metabolic, mechanical, cytokine, environment), require cancer populations to undergo greater degrees of adaptation to survive; in our approach, this greater degree of adaptation comes with a greater penalty. Consequently, we relate large vs. small local penalty terms $\beta$ to *anti-tumor* vs. *pro-tumor* IMEs. Conceptually, the baseline state (infiltrated anti-tumor IME) may give rise to three alternative

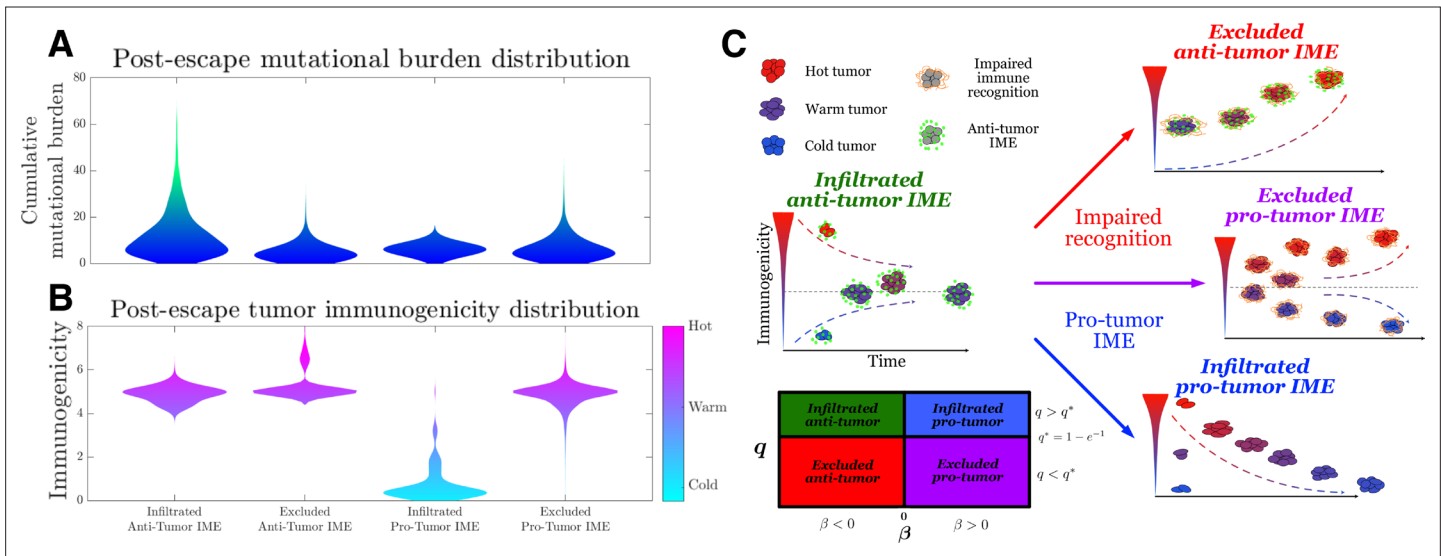

**Figure 5.** Active Evader dynamics. Violin plots of the distribution of post-immune escape. (**A**) Cumulative mutation burden. (**B**) Post-escape immunogenicity (available tumor-associated antigens [TAAs]) as a function of time for a variety of tumor immune microenvironment (IME) conditions. (Anti-tumor-infiltrated: $q = q^* + 0.1$, $\beta = 0.529$; anti-tumor-excluded: $q = q^* - 0.1$, $\beta = 0.505$; pro-tumor-infiltrated: $q = q^* + 0.1$, $\beta = -0.529$; pro-tumor-excluded: $q = q^* - 0.1$, $\beta = -0.505$. In all cases, $\beta$ chosen to give $|s_\infty| = 3$ [$s_\infty$ for the pro-tumor-infiltrated case] giving strictly positive penalties. Simulations were run until $n = 10^6$ escape events occurred for each case.) (**C**) The number of recognizable TAAs over time along with equilibrium states is depicted assuming (left) anti-tumor IME, $\beta > 0$, and efficient immune recognition. Compromises in (top right) recognition, $q < q^*$; (bottom right) the establishment of a pro-tumor IME, $\beta < 0$, or (middle right) both affect the predicted dynamical behavior of tumor immunogenicity. A phase plot partitions each case as a function of relevant critical parameter values.

The online version of this article includes the following source data and figure supplement(s) for figure 5:

**Source data 1.** Source data contains a spreadsheet of data for *Figure 5A, B*.

**Figure supplement 1.** Linear dynamics.

**Figure supplement 2.** Transition dynamics.

**Figure supplement 3.** Escape dynamics.

**Figure supplement 4.** Active evasion summary.

states (excluded anti-tumor IME, infiltrated pro-tumor IME, or excluded pro-tumor IME), based on progression.

Toward this end, we simulate the TEAL model under the above conditions and record post-escape TAA distributions. As already explained, our results predict that infiltrated ($q > q^*$) environments lead to an absorbing equilibrium state in the intervening period prior to escape, while exclusion ($q < q^*$) results in unstable equilibria. Interestingly, the sign of this equilibrium, and hence the long-term immunogenic trajectory, depends on the sign of $\beta$ (*Equations 88 and 89*). The baseline infiltrated anti-tumor case ($q > q^*$, $\beta > 0$) yields a positive and stable, mean-reverting TAA steady state, generating immunogenically 'warm' tumors. Excluded anti-tumor IMEs ($q < q^*$, $\beta > 0$) exhibit low recognition and large TAAs arrival, resulting in a unstable TAA steady state that leads to increased immunogenicity over time, resulting in 'hot' tumors. Furthermore, the infiltrated pro-tumor ($q > q^*$, $\beta < 0$) case demonstrates preserved recognition with low TAAs arrival and generates an unphysiological negative stable steady state, thereby predicting that trajectories reduce immunogenicity to zero over time, yielding 'cold' tumors. Lastly, excluded pro-tumor IMEs ($q < q^*$, $\beta < 0$), having compromises in both recognition and TAA arrival rate, result in an unstable state, above which trajectories accumulate additional TAAs over time, becoming immunogenically 'hot,' and below which the populations are predicted to reduce the number of recognizable TAAs over time, becoming 'cold' (*Figure 5A and B*). Substantial heterogeneity in the distributions of escape time predict sustained interactions in the unimpaired case (*Figure 5—figure supplement 3*). Tumor exclusion leads to hot tumors so that escape, should it occur, must do so on average prior to the accumulation of many TAAs. Conversely, pro-tumor IME with immune recognition drives TAA depletion, so escape occurs relatively early. These results are summarized in *Figure 5C*.

## Discussion

The underlying evolutionary dynamics of adaptive populations lies at the heart of many important clinical challenges, including antibiotic resistance, acquired drug resistance, immunotherapy failure, and tumor immune escape. Quantitative analytic modeling will continue to provide improved insight into these complex issues by generating fast and affordable predictions and a convenient theoretical framework for hypothesis testing. To date, virtually all of the current models of cancer evolution and the tumor-immune interaction have assumed passive acquired evolution without allowing the tumor to sense and optimally respond to the current fitness landscape in order to maximize future survival. The 'optimal escape hypothesis' is, in our opinion, worth exploring in light of the myriad examples of treatment failure and adaptive resistance.

Our analysis centered on the ability of cancer populations to adaptively respond to a measured immune state, and we have primarily focused on studying subsequent mutations resulting in the disruption of existing (targeted) tumor-associated antigenic targets and on the generation of new ones. It is important to note that independent empirical observations support the ability of cancer cells to sense their IME, and perhaps even the level of CD8+ killing that occurs therein. At the signaling level, IL-6 secreted by CTLs, macrophages, and dendritic cells in response to immune recognition has been shown to directly activate ataxia-telangiectasia mutated (ATM), a factor implicated in response to DNA damage, and this has been associated with increased metastasis and multi-drug resistance in lung cancer (*Jiang et al., 2015*; *Yan et al., 2014*). IFN-gamma released by activated CD8+ tumor-infiltrating lymphocytes activates the cell-intrinsic STING pathway in response to DNA damage in cancer, implicating an altered TME from activated CD8+ T cells that is measurable by the cancer (*Xiong et al., 2022*). Lastly, at the level of individual TCR interactions with recognized tumor cells, granzyme B release has been directly linked to DNA damage and associated CHK2 and p53 stress responses, and studies have demonstrated hSMG-1 stress-activated proteins upregulated in cancer cells following granzyme B treatment (*Meslin et al., 2011*). Moreover, granzyme release in the microenvironment serves a signaling molecule promoting a pro-inflammatory response from other immune cells (*Cullen et al., 2010*). The relatively acute response and short half-lives of downstream effectors (e.g., minutes for p53 and hours for CHK1) provide a tunable response based on the current level of immune targeting through stress-induced mutagenesis (*Bindra et al., 2007*; *Rosenberg, 2001*; *Rosenberg and Queitsch, 2014*) that in our analysis directly influences tumor-associated antigen availability.

Toward this end, we propose and analyze the TEAL model for studying and comparing passive and optimal escape mechanisms in the tumor-immune interaction. We focused our dynamic programming approach on a particular set of relations to provide analytical insight into this process. We do note, however, that the Bellman function approach to dynamic programming can be numerically implemented to obtain solutions for arbitrary functional forms of the penalty function, thereby enabling analysis of more complex assumptions where analytic progress becomes intractable. As expected, threats adopting optimal evasion strategies largely outperform their passive counterparts by increasing the rate of immune escape over prolonged cycles of cancer-immune co-evolution. In the setting of the tumor-immune interaction, the resulting TAAs available for targeting, a proxy for clinical post-detection immunotherapeutic efficacy, are augmented when cancer populations accrue large penalties for evasion and, perhaps surprisingly, when immune recognition is impaired.

Evasion dynamics of passive and active evaders are similar in some ways while different in others. Similarities include the mean-reverting stationary dynamics of both strategies under efficient immune recognition. However, the TEAL model predicts, for adaptive threats in an excluded pro-tumor IME, the emergence of an unstable state, resulting in either accrual or depletion of TAAs in a manner that depends on the current TAA abundance. This splitting behavior into 'hot' and 'cold' tumors offers insight into the microenvironmental features generating spatial immunogenic diversity within solid tumors and is consistent with prior observations (*Huss et al., 2021*; *Jia et al., 2022*; *Meiller et al., 2021*; *Lakatos et al., 2020*). This argues that TAA-depleted tumors share in common the tendency for their evasion strategies to incur less antigenic penalties. Our results suggest the possibility of altering the tumor IME to increase the immunogenicity of immune-cold tumors by making evasion more costly in a manner reminiscent of mutational meltdown (*Gabriel et al., 1993*). We remark that these dynamics are worth considering in the case of adoptive T cell-based immunotherapies, marked by their potential for exerting substantial co-evolutionary pressure on a developing malignancy (*George and Levine, 2021*). We also predict that impaired immune recognition leads to TAA accumulation, consistent with experimental observations in lung cancer wherein patients with HLA loss of heterozygosity harbored larger mutational burdens, an indirect measure of TAAs of our model (*McGranahan and Swanton, 2017*). Lastly, active evader variable mutation rates also distinguish this case from passive evaders with fixed mutation rates, and this feature is analogous to that observed in bacterial colonies faced with antibiotic selective pressure (*Windels et al., 2019*).

More generally, the TEAL framework provides a mechanistic basis for several empirical observations. First, our results would suggest that the lower observed TAA availability of hematological malignancies vs. immune-protected solid tumors, such as melanoma (*Lawrence et al., 2013*), occurs as a result of greater immune accessibility and possible immunoediting of liquid cancers. Second, our model predicts enhanced immune interactions, both natural and treatment-derived, resulting from increasing the cost of immune evasion in the evading cancer population in order to enrich the TAAs following escape. This supports the utility of neo-adjuvant radiation therapy (*McGranahan et al., 2016*) or chemotherapy (*Mouw et al., 2017*) in inducing immunogenicity. Orthogonal efforts to quantify cancer evolution have similarly predicted the benefit of larger evasion rates resulting in mutational meltdown (*McFarland et al., 2014*). Integrated together, the TEAL model can predict the balance of generated TAAs given the relative influences of recognition and evasion penalty.

Tumor antigen depletion is a concerning consequence of immunotherapy since increased recognition is desirable and required for tumor elimination. In solid tumors, one contributor to this problem is T cell exclusion (*Pai et al., 2020*). However, should effective treatment and robust tumor recognition lead to relapse, the resulting tumor has a greater chance of being TAA-depleted (*Rosenthal et al., 2019*). Other strategies that fall in this group include those that effectively reduce recognition, like the presence of T-regulatory cells. Our results suggest that this detrimental effect of targeting can be offset by increasing the 'hostility of the IME.' Strategies encourage making tumor adaptation more penalizing, such as fostering an anti-tumor environment by, for example, M1 macrophage polarization, or the inactivation of tumor-associated macrophages (*Liu et al., 2021*; *Goswami et al., 2017*).

Of course, this foundational model is not without limitation. At present, we have assumed that the recognition agent is not employing an optimized strategy informed by optimal cancer evasion. Instead, we have detailed our results for arbitrarily imputed recognition landscapes, which is useful for predicting the response of an aggressive evader like cancer to particular immunotherapeutic interventions, such as hematopoietic stem cell transplant and adoptive T cell therapy, where the clinician

has temporal control over treatment. Identification of such optimal treatment strategies upon quantification of disease evasion aggressiveness is of paramount importance. In this foundational model, we demonstrated the dynamics of immune recognition of an adaptive population of cancer cells expressing a purely clonal pattern of antigens. Our model implicitly equates antigen loss and the progression of a subpopulation currently adapted to evade immune targeting – either by direct pruning of the fittest subclone or by stochastic emergence and subsequent growth of a new one lacking the targeted antigens – as equivalent. Here, we tracked the fittest clone represented by a core set clonal antigens. We remark that heterogeneous populations each having a distinct subclonal signature can also be tracked, but the corresponding antigen-driven selection and fitness cost to each clone would be coupled through shared antigens (see 'Methods'). Finally, we note that this extended approach implicitly assumes that antigen detection rates over a given period are subclone size-independent, given that antigens are tracked over a period where each of the clones with comparable fitness would be detectable by the immune system during their growth trajectory en route to attempted escape.

Lastly, cancers characterized by co-evolutionary dynamics resulting in large variability in population size prior to escape or elimination would require in general that recognition and evasion parameters depend on the current period. While possible to incorporate, we have for foundational understanding assumed these to be constant. In this discrete-time evolutionary model, the intertemporal period considered represents the time period between the earliest moment that the adaptive immune system may identify a cancer clone and the latest point after which such a recognition event would no longer be able to prevent cancer escape (*George and Levine, 2020*). This effectively gives $q$ a probabilistic representation for the total rate of opportunity to recognize a given TAA during cancer progression. Implementing this model in cancer subtype-specific contexts thus requires a consideration of the per-cell division rates, for example.

We detailed strategies that affect the number of TAAs present following escape. In addition to quantity, variations in individual TAA antigenicity could affect overall immunogenicity, but we do not as yet take this into account. In future work, individual antigenicities could be built in by allowing individual TAA contributions to $s_n$ and $q$ to depend on the particular TAA. Many additional features contribute to the immune landscape. Here, we focused on TAA availability and effects of general immune recognition rates and IME hostility on TAA accrual. Future efforts may incorporate additional cancer-specific features, including antigen presentation, immunomodulatory gene expression, and measured immune signatures present in the IME.

These optimized dynamics are proposed in the absence of the precise mechanistic details of cancer decision-making. Further studies linking changes in the evasion rates to cell signaling are necessary next steps at elucidating a possible mechanism of optimal evasion. Our framework serves as a tool for evaluating the extent of evasion aggressiveness in a variety of observed disease contexts, including cancer. Differentiating dynamics of passive and adaptive evasion mechanisms is a first step to understanding this difference, its importance underscored by the large implications such an understanding would have on our approach to treatment.

The TEAL model represents a framework broadly applicable for studying population behavior consistent with optimized collective decision-making, and subsequent experimental validation or refutation is of highest priority. Future direction aims to apply this framework for personalizing optimal interventions that maximize disease elimination probabilities. Consequently, stochastic analysis and optimal control theory are indispensable tools for better understanding the complex cancer-immune interaction. Defeating an evolving cancer population has provided a persistent challenge to researchers and clinicians, with the majority of progress heralded by fundamental discoveries on cancer behavior, and additional insights require a more detailed understanding of cancer evasion. The possibility that cancer population-level strategies are somewhat informed to the present recognition threat would have a radical effect on our own optimal treatment approach.

## Methods
### Passive evader in an adaptive environment

Let $\mathcal{S}_n$ denote the set of tumor antigens recognizable by the immune system and present at period $n$ on a population of cancer cells, and let $s_n = |\mathcal{S}_n|$ count their number ($|\mathcal{A}|$ denotes the cardinality of set $\mathcal{A}$). From one period to the next, each of the $s_n$ detectable antigens may be independently and

identically detected by the immune system with probability $q$ per antigen. We let $\mathcal{R}_n \subseteq \mathcal{S}_n$ denote the collection of antigens that are recognized by the immune system at time $n$. As the immune system targets and begins to eliminate cells via the $\mathcal{R}_n$ antigens, the cancer population has an opportunity to lose or downregulate each of the $r_n = |\mathcal{R}_n|$ recognized antigens with a similar independent and identical manner. The rate of antigen loss $\pi_n$ may in general vary as a function of time and environmental features (considered in Section Active evader in an adaptive environment). In this section, we assume it is passively fixed and denote this rate as $p$. We denote the collection of antigens that are lost by the cancer population at time $n$ by $\mathcal{L}_n \subseteq \mathcal{S}_n$. We track the number of recognized and lost antigens at time $n$ by $r_n$ and $\ell_n = |\mathcal{L}_n|$, respectively, so that $\ell_n \leq r_n \leq s_n$.

The system evolves as follows (*Figure 1—figure supplements 1 and 2*): If $\mathcal{R}_n = \emptyset$, then the immune system is unable to recognize any tumor antigen at time $n$ and so the process ends in cancer escape. Since in this case the immune system *loses*, we denote this event by $L_n$. If $\mathcal{R}_n \neq \emptyset$, then the immune system recognizes the threat by at least one TAA and one of two outcomes results: The first possibility is that the cancer population successfully downregulates or loses all of the targeted antigens, expressed as $\mathcal{L}_n = \mathcal{R}_n$, and survives to the next time step. We call this a tie and denote the event by $E_n$. Alternatively, the cancer population is unable to lose every recognized antigen and subsequently becomes eliminated. This means the immune system has *won* so we denote this event by $W_n$. Although the recognition and evasion probabilities may in general be clonally and temporally dependent, we assume fixed probabilities for the recognition, $q$, and evasion, $p$, of individual antigens. In the event of a tie, $s_n - r_n$ antigens remain, with the addition of a basal antigen arrival rate $\beta$ and a possibly noisy penalty term $f_n$ to reflect the production of new antigens as the population evolves. For simplicity, we assume the $\beta$ to be constant and the $f_n$ a sequence of independent, identically distributed (IID) random variables with mean $f$. While it is in general possible that the distributions of $r_n$ and $\ell_n$ be both state- and time-dependent, we focus on the foundational example above.

This process is identical to the following game between two players, hereafter referred to as the 'Recognizer' (immune system) and the 'Evader' (threat): the Recognizer starts off with a collection, $\mathcal{S}_0$, of $s_0$ coins and begins her turn by flipping each coin with IID success probability $q$. If she has no success ($\mathcal{R}_0 = 0$), she loses (denoted by event $L_0$) and the game ends. If $r_0 > 0$ of her coins land on heads, then the next turn goes to the Evader, who proceeds to flip his $r_0$ coins with IID success probability $p$ in an attempt to match the Recognizer's successful coin flips. The Evader must succeed in all coin flips ($\mathcal{L}_0 = \mathcal{R}_0$) for the turn to end in a tie (equilibrium between Evader and Recognizer), given by event $E_0$. Otherwise, he loses and the game ends with a Recognizer win, (event $W_0$). If a tie occurs then both players restart the game, but only after the removal from $\mathcal{S}_0$ of the $r_0$ coins that landed on heads for both players as well as the addition of a random number $f_0$ of new coins. The Evader wins by default if a new turn begins and there are no longer any remaining coins to flip.

## Probability of equilibrium

It is immediately apparent that this game is unfair to the Evader if $s_0$ is much larger than 1, unless the recognition probability $q$ is low and the evasion probability $p$ is high. We motivate the following analysis with this in mind and proceed to characterize the dynamics of this stochastic process. Clearly, the number of recognized and lost antigens during each period is binomially distributed, their respective distributions given by

$$r_n \sim \mathrm{Binom}(s_n, q); \qquad \ell_n \sim \mathrm{Binom}(r_n, p). \tag{12}$$

The event that the immune and cancer systems are in equilibrium (non-escape and non-extinction) may be written as

$$E_n = [\mathcal{L}_n = \mathcal{R}_n \supsetneq \emptyset] = [\ell_n = r_n > 0]. \tag{13}$$

One might expect that the number of antigens lost at time $n$ is affected by knowledge of whether or not the game continues to be played. The distribution of $\ell_n$ conditioned on equilibrium may be characterized by conditioning on the number of recognized antigens at time $n$. To this end, let $F_{n,r} = [r_n = r]$ denote the event that $r$ antigens are recognized at period $n$, with

$$\mathbb{P}\left(F_{n,r}\right) = \binom{s_n}{r} q^r (1-q)^{s_n - r}. \tag{14}$$

We remark that events $\{F_{n,r}\}_r$ are disjoint and exhaustive; in other words, for sample space $\Omega$,

$$\bigcup_{r=0}^{s_n} F_{n,r} = \Omega; \qquad F_{n,i} \cap F_{n,j} = \emptyset, \quad \text{for } i \neq j. \tag{15}$$

Additionally, we note that equilibrium cannot occur if no antigens are recognized (i.e., $F_{n,0} = [\mathcal{R}_n = \emptyset]$). Lastly,

$$\mathbb{P}\left(E_n \mid F_{n,r}\right) = p^r, \tag{16}$$

since if $r$ antigens are recognized then $\mathcal{L}_n = \mathcal{R}_n$ occurs if and only if each of the $l_n = r_n$ recognition positions are exactly matched with $r_n$ evasions. We will make use of the following variables to simplify subsequent results:

$$\eta \equiv (1-q) + qp = \left[1 - q(1-p)\right]; \qquad \gamma \equiv 1 - q. \tag{17}$$

Here, $\eta$ may be interpreted as the probability of the complement of the following event: 'recognition occurs without matched evasion for a single antigen.' In other words, $\eta$ is the probability that equilibrium exists at one antigen position provided that there is at least one available antigen for immune targeting. This event occurs in one of two disjoint ways for a single antigen: either there is no recognition, and so equilibrium occurs regardless of evasion, or there is recognition that must also be matched by evasion. The joint distribution of recognized and lost antigens is given by the probability mass function

$$\begin{aligned}
m(r, l) &= \mathbb{P}\left([r_n = r] \cap [\ell_n = \ell]\right) \\
&= \mathbb{P}\left(\ell_n = \ell \mid r_n = r\right) \mathbb{P}\left(r_n = r\right) \\
&= \binom{r}{\ell} p^\ell (1-p)^{r-\ell} \cdot \binom{s_n}{r} q^r (1-q)^{s_n - r}.
\end{aligned} \tag{18}$$

The probability that equilibrium occurs and the process continues at period $n$ is given by

$$\begin{aligned}
\mathbb{P}\left(E_n\right) &= \sum_{r=1}^{s_n} m(r, r) \\
&= \sum_{r=1}^{s_n} \binom{s_n}{r} (pq)^r (1-q)^{s_n - r} \\
&= (1-q)^{s_n} \left[ \left( \frac{q - pq - 1}{q - 1} \right)^{s_n} - 1 \right] \\
&= \left[ 1 - q(1-p) \right]^{s_n} - (1-q)^{s_n} \\
&= \eta^{s_n} - \gamma^{s_n},
\end{aligned} \tag{19}$$

which is equal to the probability of equilibrium occurring at every position minus the probability that all of the $s_n$ antigens are not recognized, since at least one recognized antigen is required for equilibrium to occur.

## Break-even probability

The process is usually more favorable for the Recognizer. The Recognizer loses at period $n$ if there are zero recognition events, and this occurs with probability

$$\mathbb{P}\left(L_n\right) = \gamma^{s_n}. \tag{20}$$

The Recognizer wins at period $n$ if she does not lose or tie, which occurs with probability

$$\mathbb{P}\left(W_n\right) = 1 - \left( \mathbb{P}\left(E_n\right) + \mathbb{P}\left(L_n\right) \right) = 1 - \eta^{s_n}. \tag{21}$$

If $q$ and $s_n$ are given, then the evasion probability $p$ required for equal probabilities of Recognizer failure and success, or the *break-even probability*, is given by

$$p_{\text{even}} = \frac{(1 - \gamma^s)^{1/s} - \gamma}{1 - \gamma}, \tag{22}$$

and exists whenever $p_{\text{even}} > 0$. We plot $p_{\text{even}}$ as a function of recognition probability $q$ for various numbers of TAAs, $s$ (**Figure 1—figure supplement 5A**). The 'fair-game' line indicates where the break-even evasion probability is always equal to the recognition probability. Regions where the break-even probability localizes above the fair-game line favor the Recognizer since there the evasion rates $p$ must be higher than recognition rates $q$ for the game to be fair. Alternatively, areas below the break-even curve favor the Evader. It is clear from **Figure 1—figure supplement 5B** that the process favors recognition for a majority of parameter choices $(p, q)$ in all cases except for when $s = 1$. Thus, the process is largely unfair and mostly favors the Recognizer over the Evader when $p = q$ so long as $s$ is not small. In order for the Evader to have a reasonable chance of success, either the evasion probability must be very large or the number of TAAs must remain small.

## Tracking distinct clones

The above describes a clonal population harboring a core minimal set of TAAs for which recognition and downregulation ultimately determine cancer escape, elimination, or equilibrium. Our model can however be adapted to study the more general scenario involving a clonal hierarchy of heterogeneous cancer cells. We illustrate this by considering a population of cells with a set $C$ of $c = |C|$ core clonal TAAs, together with distinct groups of cells with subclonal collections of TAAs $S_1$ and $S_2$ (having size $s_1 = |S_1|$ and $s_2 = |S_2|$, respectively). The relevant populations therefore have antigen sets given by $P_1 = C \cup S_1$ and $P_2 = C \cup S_2$. The basic event considered in the foundational model, $[r_n > 0]$, must now be replaced by the event that recognition occurs in both $P_1$ and $P_2$; in the absence of recognition of both subclones, the cancer escapes. Recognition happens either if there is a recognition event $r$ in $C$ or if there are simultaneous recognition events $r_1$ in $S_1$ and $r_2$ in $S_2$. Assuming that TAA recognition occurs independently as before with probability $q$, the total probability of relevant recognition, originally $(1 - \gamma^{s_n})$, is now given by $(1 - \gamma^c) + \gamma^c(1 - \gamma^{r_1})(1 - \gamma^{r_2})$. The first term characterizes the coupling of the fate of both subclones should a common TAA be recognized, while the latter term represents the parallel recognition process required to control each subclone separately via subclonal TAA recognition. Lastly, assuming that recognition proceeds either by a shared TAA in $C$ or instead by subclonal TAAs in both $S_1$ and $S_2$, then the probability of elimination and progression proceed identically as before. In the remainder of the discussion, we will, for baseline understanding, only track a core set of clonal antigens on the fittest clone.

## Distribution of lost antigens

The process transitions at period $n$ if and only if equilibrium occurs, which means that the number of lost antigens match those recognized and are strictly positive. In other words,

$$E_n = \left[ \ell_n = r_n > 0 \right]. \tag{23}$$

The survival probability as a function of $q$ and $p$ are plotted for various choices of $s$ in **Figure 1—figure supplement 6**. From this, we find that equilibrium occurs with high probability for large evasion rates, $p$, as well as for recognition rates $q$ that vary inversely with the number of recognizable antigens. This coincides with conditions that do not disadvantage the Evader so that the equilibrium probability is maintained. We remark that recognition and evasion rates in general vary with the IME. We shall subsequently restrict our attention to large recognition probabilities ($p > 1/2$).

## Exact dynamics

Let $I_F$ denote the usual indicator random variable on event $F$:

$$I_{F(\omega)} = \begin{cases} 1, & \omega \in F; \\ 0, & \omega \notin F. \end{cases} \tag{24}$$

If $r_n$ is unknown, then the distribution of $\ell_n$ follows that of $r_n$ on a strictly positive outcome normalized to the probability of surviving:

$$\begin{aligned}
\mathbb{P}\left(\ell_n = \ell \mid E_n\right) &= \mathbb{P}\left(\left[\ell_n = \ell\right] \cap \left[\ell_n = r_n > 0\right]\right) / \mathbb{P}\left(E_n\right) \\
&= \mathbb{P}\left(r_n = \ell_n = \ell > 0\right) / \left(\eta^{s_n} - \gamma^{s_n}\right) \\
&= \begin{cases} m(\ell, \ell)/\mathbb{P}\left(E_n\right), & 0 < \ell \le s_n; \\ 0, & \ell = 0. \end{cases} \\
&= I_{[\ell > 0]} \binom{s_n}{\ell} \left[p(1-\gamma)\right]^\ell \frac{\gamma^{s_n - \ell}}{\left(\eta^{s_n} - \gamma^{s_n}\right)}.
\end{aligned}$$
(25)

In this case, the mean number of lost antigens conditioned on a tie becomes

$$\begin{aligned}
\mathbb{E}\left[\ell_n \mid E_n\right] &= \sum_{\ell=0}^{s_n} \ell\, \mathbb{P}\left(\ell_n = \ell \mid E_n\right) \\
&= \left(\eta^{s_n} - \gamma^{s_n}\right)^{-1} \sum_{\ell=1}^{s_n} \ell \binom{s_n}{\ell} \left[p(1-\gamma)\right]^\ell \gamma^{s-\ell} \\
&= \frac{p(1-\gamma)\eta^{s_n - 1}}{\eta^{s_n} - \gamma^{s_n}} s_n.
\end{aligned}$$
(26)

Of course, for any realized number of recognized antigens $r_n$ at period $n$ (event $F_{n,r} = [r_n = r]$), the number of lost antigens conditional on equilibrium $\ell_n$ is completely determined since

$$\begin{aligned}
\mathbb{P}\left(\ell_n = \ell \mid E_n \cap F_{n,r}\right) &= \mathbb{P}\left(\ell_n = \ell \mid \ell_n = r_n = r > 0\right) \\
&= I_{[\ell = r]},
\end{aligned}$$
(27)

so that the conditional mean number of lost antigens must match exactly those recognized:

$$\begin{aligned}
\mathbb{E}\left[\ell_n \mid E_n \cap F_{n,r}\right] &= \sum_{\ell=0}^{s_n} \ell \cdot \mathbb{P}\left(\ell_n = \ell \mid E_n \cap F_{n,r}\right) \\
&= \sum_{\ell=0}^{s_n} \ell I_{[\ell = r]} = r.
\end{aligned}$$
(28)

### Mean transition behavior

The state transition equation for this process is given by **Equation 1**:

$$s_{n+1} = s_n - \ell_n + \beta + f_n,$$

where $\beta + f_n$ represents the arrival of new antigens through a basal production rate $\beta$ plus additional antigens $\{f_n\}_n$ that possibly depend on the evasion strategy employed. In our model, we will assume that the $\{f_n\}_n$ are IID random penalties with mean $\mathbb{E}\left[f_n\right] = f$ and finite variance (e.g., Poisson-distributed). Given this, we will now characterize the mean transition behavior conditioned on equilibrium and the information available at the present moment. We write $\mathbb{E}_n\left[\cdot\right]$ to denote the conditional expectation with respect to date-$n$ information.

### Exact dynamics

The mean number of detectable antigens evolves according to the difference equation (**Equation 3**):

$$\begin{aligned}
\mathbb{E}_n\left[s_{n+1} \mid E_n\right] &= \mathbb{E}_n\left[s_n - \ell_n + \beta + f_n \mid E_n\right] \\
&= \mathbb{E}_n\left[s_n\right] - \mathbb{E}_n\left[\ell_n \mid E_n\right] + \beta + \mathbb{E}\left[f_n\right] \\
&= s_n - \frac{p(1-\gamma)\eta^{s_n - 1}}{\eta^{s_n} - \gamma^{s_n}} s_n + (\beta + f),
\end{aligned}$$

which gives **Equation 3** and follows since $s_n$ is measurable at period $n$ and independent from $E_n$, while $f_n$ is independent from period $n$ and $E_n$. This process is mean stationary at $s_n = \mu$ whenever

$$\Delta s_n \equiv \mathbb{E}_n\left[s_{n+1} \mid E_n\right] - s_n = 0$$
(29)

giving

$$\mu = \left(\frac{\beta + f}{q}\right)\left(\frac{\eta^\mu - \gamma^\mu}{p\eta^{\mu-1}}\right).$$
(30)

Plots of fixed points of *Equation 3* are illustrated in *Figure 1—figure supplement 7* for $p > 1/2$ and $q$ away from zero for small total mean antigen accumulation rates $\beta + f$. As expected, increases in $(\beta + f)$ result in higher equilibria. In the large $\mathbb{P}(E_n)$ region of interest, increased $q$ results in a lower number of detectable antigens at equilibrium since more are recognized during each period.

## Approximate dynamics

If $r_n$ is explicitly given, then the mean transition equation simplifies to

$$
\begin{aligned}
\mathbb{E}_n\left[s_{n+1} \mid E_n \cap F_{n,r}\right] &= \mathbb{E}_n\left[s_n - \ell_n + \beta + f_n \mid E_n \cap F_{n,r}\right] \\
&= s_n - \mathbb{E}_n\left[\ell_n \mid E_n \cap F_{n,r}\right] + \beta + \mathbb{E}\left[f_n\right] \\
&= s_n - r_n + \beta + f,
\end{aligned}
\tag{31}
$$

since $s_n$ is measurable at period $n$, while $f_n$ is independent from period $n$ and $E_n \cap F_{n,r}$. We can use this to approximate the exact recognition dynamics described above by assuming $r_n = \mathbb{E}_n\left[r_n\right] = qs_n$. In this case, we have *Equation 4*:

$$
\mathbb{E}_n\left[s_{n+1} \mid E_n \cap F_{n,r}\right] = (1 - q)s_n + \beta + f.
$$

The equilibrium may be given explicitly as

$$
\tilde{\mu} = (\beta + f)/(1 - \gamma) = (\beta + f)/q.
\tag{32}
$$

We distinguish the approximate equilibrium $\tilde{\mu}$ from that of exact case $\mu$, the latter incorporating a correction term arising from the fact that knowledge of equilibrium occurring requires a larger average value of $r_n$ above $qs_n$ since equilibrium occurs only when $r_n > 0$. We remark that the steady states given by *Equations 30 and 32* are close to one another for small penalty (*Figure 1—figure supplement 8*) and parameter regions that overlap with those having large equilibrium probabilities ($p \sim 1$, $q > 0.5$; *Figure 1—figure supplement 6*), which intuitively suggests that a process driven by its mean overlaps well with one conditional on equilibrium provided the escape and elimination probabilities are small. We obtain good agreement between averages of large-scale simulations of the process, together with the predicted exact and approximate steady states for $p, q > 0.5$ and small penalty (*Figure 1—figure supplement 9*). Of course, the mean dynamics are also approximate since $qs_n$ is in general non-integer-valued. With this in mind, we focus on the dynamics given by *Equation 31*.

Here, $r_n$ is Binomially distributed conditional on the number of current antigens, so that

$$
\mathbb{E}_n\left[r_n\right] = qs_n; \qquad \mathbb{V}\text{ar}_n\left[r_n\right] = q(1 - q)s_n.
\tag{33}
$$

We define the following zero-mean noise variable

$$
\varepsilon_n \equiv (f_n - f) - (r_n - qs_n),
\tag{34}
$$

and rewrite *Equation 1* as

$$
s_{n+1} = \gamma s_n + \beta + f + \varepsilon_n.
\tag{35}
$$

This is none other than a first-order autoregressive, or AR(1), process with innovation terms $\varepsilon_n$ comprised of endogenous noise due to the variance in the number of recognized antigens and exogenous noise generated by fluctuations in the random penalty term.

The process is stable for all but trivial choices of probability $\gamma$. The mean behavior evolves according to

$$
\mathbb{E}_n\left[s_{n+1}\right] = \mathbb{E}_n\left[\gamma s_n + \beta + f - \varepsilon_n\right] = \gamma s_n + \beta + f,
\tag{36}
$$

which ultimately gives *Equation 9*:

$$
\begin{aligned}
\mathbb{E}\left[s_n\right] &= \gamma^n s_0 + (\beta + f) \sum_{j=0}^{n-1} \gamma^j \\
&= \gamma^n s_0 + \left(\frac{1 - \gamma^n}{1 - \gamma}\right)(\beta + f) \\
&\to (\beta + f)/q \quad \text{as } n \to \infty,
\end{aligned}
$$

thus showing agreement in mean with the fixed point given by **Equation 32**. Of course, $s_n = \tilde{\mu} = (\beta + f)/q$ satisfies the martingale property:

$$\mathbb{E}\left[s_{n+1}\right] = \gamma(\beta + f)/q + (\beta + f) = (\beta + f)/q = s_n, \tag{37}$$

and the process tends toward steady state with expected intertemporal difference

$$\left|\mathbb{E}\left[s_{n+1}\right] - \mathbb{E}\left[s_n\right]\right| = \gamma^n \left|(\beta + f) - qs_0\right|. \tag{38}$$

The variance at stationarity, $\mathbb{V}\mathrm{ar}\left(s_n\right)$, can be calculated by solving for the fixed point of

$$\mathbb{V}\mathrm{ar}\left(s\right) = \gamma^2 \mathbb{V}\mathrm{ar}\left(s\right) + \sigma_f^2, \tag{39}$$

giving

$$\mathbb{V}\mathrm{ar}\left(s_n\right) = \sigma_f^2/(1 - \gamma^2). \tag{40}$$

## Recognizer success probability

For the event $W_n$ (resp. $L_n$) that the Recognizer wins (resp. loses) at period $n$, and for the event $E_n$ of equilibrium at period $n$, we have

$$\mathbb{P}\left(W_n\right) = \mathbb{P}\left(E_{n-1}\right)\left(1 - \eta^{s_n}\right) \tag{41}$$

$$\mathbb{P}\left(L_n\right) = \mathbb{P}\left(E_{n-1}\right)\gamma^{s_n}, \tag{42}$$

$$\mathbb{P}\left(E_n\right) = \mathbb{P}\left(E_{n-1}\right)\left(\eta^{s_n} - \gamma^{s_n}\right). \tag{43}$$

These relationships, along with the implicit evolution given by **Equation 32**, are used to approximate ultimate Recognizer success probabilities for all possible $p$ and $q$ against several choices of initial antigen number $s_0$ and mean antigen arrival rate $\beta + f$, and are compared with simulations of using actual transitions via **Equation 29** (**Figure 1—figure supplement 10**). We find good agreement between these methods in characterizing the final outcome over a variety of parameter choices, where accuracy is highest in the relevant parameter region of interest. In particular, the left column of **Figure 1—figure supplement 10** details the likelihood that a (static) threat is controlled in the special case where no penalty is assumed.

## Mutation accumulation rate and tumor antigen availability

The above analysis was motivated by a desire to explain both genetic and non-genetic possibilities leading to recognition evasion. We can consider applying this model to strictly describe genetic evasion in the form of somatic mutations leading either to the generation of (recognizable) tumor-associated antigens or to escape via the removal of these antigens. Using the above framework, mutations, denoted by $\lambda$, accumulate across each period in proportion to the sum of antigens down-regulated to enhance escape and antigens gained via basal arrival and penalty. Thus their rate of accumulation may be expressed by

$$\nu(n) \equiv \frac{\Delta\lambda(n)}{\Delta n} \propto \ell_n + \beta + f_n. \tag{44}$$

Together with the fact that $\ell_n = r_n$ during progression, we have for the mean rate of mutant accumulation

$$\mathbb{E}\left[\nu(n)\right] \propto \mathbb{E}\left[\mathbb{E}\left[r_n \mid s_n\right] + (\beta + f)\right]$$
$$= q\mathbb{E}\left[s_n\right] + \beta + f \tag{45}$$
$$\to 2(\beta + f) \quad \text{as} \quad n \to \infty,$$

ultimately giving

$$\lambda(n) \propto 2(\beta + f)n. \tag{46}$$

which predicts that the rate of mutational acquisition is linear in time, consistent with empirical observation (*Alexandrov et al., 2013*; *Lawrence et al., 2013*). Heuristically, tumors that survive while accumulating an average of $\beta + f$ targetable alterations must balance those gains by $\beta + f$ additional evasion events. This theory predicts, perhaps surprisingly, that the mutation rate is a direct reflection of the penalty paid for cancer progression as a function of the basal antigen arrival rate and contributions from the local environment. Tumors having a more difficult time surviving in a hostile or restrictive environment would be predicted to have higher rates of mutation. In this context, high mutational signatures are predicted to be correlated with tumors that are more susceptible to recognition. For a passive Evader, our theory predicts that the observed mutation rate depends only on basal arrival and mean penalty term for cancer progression, unaffected by recognition rate. On the other hand, the stationary number of available antigens, approximated by $\tilde{\mu} = (\beta + f)/q$, varies directly with evasion penalty and inversely with antigen recognition rate. Moreover, mutation or adaptation accumulation is expected to converge to a stable steady state for all allowable recognition, evasion, and penalty rates.

## Active evader in an adaptive environment

In the previous section, we considered the predicted dynamical behavior when the Evader is assumed to adopt a fixed strategy. In that case, if number of detectable antigens is moderately large ($s_0 \sim 10$), then the game is biased against the Evader for most combinations of evasion and recognition success probabilities (Section Break-even probability). Additionally, mean transitions in the number of recognizable antigens obey an AR(1) process tending toward the quotient of the mean penalty and recognition rate (Section Mean transition behavior). Moreover, this behavior predicts that the observed mutation accumulation rate is linear in time and proportional to the sum of basal antigen creation rate and mean penalty term (Section Mutation accumulation rate and tumor antigen availability). Here, we allow for the Evader to optimally select his evasion rate $\pi_n$ at each period (*Figure 1—figure supplement 3*). Larger success rates come at the cost of adding back more recognition opportunities in the subsequent time step, so that the Evader employs a strategy to maximize his survival or likelihood of escape. This framework is motivated by the observation that cancer threats are known to accumulate perhaps mildly deleterious mutations that occur passively during evolution to obtain rare 'driver' mutations (*McFarland et al., 2014*). The novelty here is that we propose a unifying theoretical framework to investigate the resulting strategy employed by a cancer population if the choice of evasion is planned based on knowledge of the current antigen landscape and hostility, or number of recognized targets.

In contrast with the prior section, which considered temporal evolution as a function of fixed evasion rate $p$ and random penalty $f_n$, here, the evasion rate $\pi_n$ may depend on time, and for simplicity we consider deterministic penalties. In order to properly frame this problem in a manner suitable to handle via dynamic programming, we define the necessary parameters, expectation, and value functions below. We assume that the process evolves according to state transition equation,

$$s_{n+1} = s_n - r_n + \beta + f_n, \tag{47}$$

and that conditional expectations are taken with respect to $\mathcal{F}_n$, the natural filtration (*Karatzas and Shreve, 1998*) with respect to the underlying process.

If at time $n$ knowledge of total $s_n$ and recognized $r_n$ targets is known, then the Evader's objective is to select a policy $\pi \equiv \{\pi_n, \pi_{n+1}, \dots\}$ that maximizes the sum of present and future rewards, $R(s_n, r_n, \pi_n)$, which in general depend on the current state, $s_n$, as well as the Recognizer, $r_n$, and Evader, $\pi_n$, actions. The value function is defined to be the maximal attainable sum of expected future rewards, given by

$$J_n(s_n) = \sup_{\pi} \mathbb{E}_n \left[ \sum_{m=n}^{\infty} R\left(s_m, r_m, \pi_m\right) \right]. \tag{48}$$

Problems that may be framed in this context have been well-studied and utilize a rich theory of stochastic dynamic programming, originally proposed by *Bellman, 1954*; *Bellman and Dreyfus, 1959*. Bellman's Principle of Optimality and Bellman equation for a stationary solution (independent of starting time) are given via backward induction by

$$J(s_n) = R(s_n, r_n, \pi_n) + J(s_{n+1}). \tag{49}$$

*Equation 49* states that the maximal attainable value at period $n$ is given by the sum of the maximal attainable value at the next time step, $J(s_{n+1})$, and the $n$-period reward of strategy $\pi_n$ obeying *Equation 48*. For the problem at hand, we assume that the Evader receives a normalized reward of either $R_n = 1$ if it escapes at any time period (there is no temporal discount for escape at later periods), or $R_n = 0$ if it is eliminated. In this case, we may draw a decision tree for the $n$-period problem in terms of the value function $J$, current antigen number $s_n$, Recognizer antigen recognition miss probability $\gamma = 1 - q$, number of recognized antigens $r_n$, and Evader strategy, $\pi_n$ (*Figure 1—figure supplement 4*). Here, $\pi_n$ represents the $n$-period probability of antigen loss by the Evader.

Using the dynamic programming principle, the Bellman equation under uncertainty takes the form given by *Equation 5*:

$$J(s_n) = \max_{\pi_n}\left\{ \mathbb{E}_n\left[\pi_n^{r_n}\left[\gamma^{s_{n+1}} + (1 - \gamma^{s_{n+1}})J(s_{n+1})\right]\right]\right\}.$$

Under a particular choice of assumed penalty and transition equation, we can calculate an exact, closed-form solution to the dynamic program in *Equation 5*. This solution generates an optimal policy, given by $\pi^* = \{\pi_1^*, \pi_2^*, \ldots, \pi_n^*, \ldots\}$, a sequence of optimal decisions, in addition to the maximal value at each time assuming the optimal policy, given by $J(s_n)|_{\pi_n^*}$.

## Constitutive relations for intertemporal penalty

We make the following assumptions in our setting to make this problem more tractable. The first assumption is that the penalty function is time-homogeneous and deterministic:

$$f(s_n, r_n, \pi_n), \quad \pi_n \in [0, 1], \quad s_n, r_n \in \mathbb{Z}^+. \tag{50}$$

Conditional on progressing to the next period, the transition equation takes the following form:

$$s_{n+1} = s_n - r_n + \beta + f(s_n, r_n, \pi_n). \tag{51}$$

In cases where we wish to emphasize the dependence of the transition equation on $\pi_n$, we will denote $s_{n+1}$ by $g(\pi_n)$ so that

$$g(s_n, \pi_n) = s_{n+1} \tag{52}$$

The second assumption is that this penalty is $\pi_n$-linear, given by *Equation 2*:

$$f(s_n, r_n, \pi_n) = h_m(s_n, r_n)\pi_n$$

for positive $h_m$.

In order to analytically characterize the solution, we assume that $r_n$ is known prior to choosing $\pi_n$ ($r_n \in \mathcal{F}_n$). In the analogous coin game, the Evader is allowed to see the success of his opponent, the Recognizer, prior to choosing a strategy. In this case, the dynamic program has a solution if we also assume that the linear penalty term can be represented by

$$h_m(s_n, r_n) = \frac{r_n}{c}\left(\frac{1}{\delta_n} \cdot \frac{1 - \gamma^{s_n}}{1 - \gamma}\right)^{1/r_n} \tag{53}$$

with $c \equiv -\ln\gamma > 0$ and $0 < \delta_n \leq 1$. This assumption implies that the marginal penalty of increasing $\pi_n$ is asymptotically proportional to the number of recognized antigens. This is reasonable to assume, for example, in cases where significant immune system recognition and tumor killing create an environment that makes subsequent adaptation more costly, resulting possibly from increased inflammation. The constant $\delta_n$, a free variable, is inversely related to aversion of the Evader strategy so that larger values imply a bolder evasion strategy for all else held constant. This parameter may in general vary temporally and as a function of disease subtype.

## Dynamic programming solution

In the above case, we may find an exact solution to the optimal programming problem. Since $r_n \in \mathcal{F}_n$ (the filtration generated by the evolution of $s_n$ and the Recognizer action at time $n$), the stationary Bellman equation takes the form

$$J(s_n) = \max_{0 \le \pi_n \le 1} \left\{ \pi_n^{r_n} \left[ \gamma^{s_{n+1}} + (1 - \gamma^{s_{n+1}}) J(s_{n+1}) \right] \right\}. \tag{54}$$

For simplicity in the subsequent definition, we drop the period index, rewriting *Equation 54* as

$$J(s) = \max_{0 \le \pi \le 1} \left\{ \pi^r \left[ \gamma^{g(s,\pi)} + (1 - \gamma^{g(s,\pi)}) J(g(s,\pi)) \right] \right\} \tag{55}$$

Using $c \equiv -\ln \gamma$, the first-order condition (FOC) is

$$\frac{\partial}{\partial \pi} \left\{ \pi^r \left[ e^{-cg(s,\pi)} + (1 - e^{-cg(s,\pi)}) J(g(s,\pi)) \right] \right\} = 0. \tag{56}$$

In expanded form, the FOC becomes

$$0 = \pi^{r-1} \left\{ r \left[ e^{-cg} + (1 - e^{-cg}) J(g) \right] \right.$$
$$\left. + \pi \left[ -c \frac{\partial g}{\partial \pi} e^{-cg} + c \frac{\partial g}{\partial \pi} e^{-cg} J(g) + (1 - e^{-cg}) \frac{\partial J}{\partial g} \frac{\partial g}{\partial \pi} \right] \right\}. \tag{57}$$

From *Equation 2*, we have that

$$\frac{\partial f}{\partial \pi} = \frac{\partial g}{\partial \pi} = h_m. \tag{58}$$

We postulate that the solution takes the form of *Equation 6*:

$$J(s) = \frac{A\gamma^s}{1 - \gamma^s}.$$

so that

$$\frac{\partial J}{\partial s} = -\frac{cJ(s)}{(1 - e^{-cs})}. \tag{59}$$

This, together with *Equation 59*, reduces *Equation 58* to

$$\pi^{r-1} \left[ e^{-cg} + (1 - e^{-cg}) J(g) \right] (r - ch_m \pi) = 0. \tag{60}$$

Thus, the optimal Evader success probability, $\pi^*$, is given by

$$\pi^* = r/ch_m. \tag{61}$$

Under Evader optimal strategy, the transition equation in *Equation 51* becomes

$$g^* \equiv g(s, \pi^*) = s - r + \beta + f(s, r, \pi^*)$$
$$= s - r + (\beta + r/c). \tag{62}$$

We next confirm that this satisfies the Bellman equation (*Equation 55*). The above solution implies

$$J(s) = \pi^{*r} \left[ \gamma^{g^*} + (1 - \gamma^{g^*}) J(g^*) \right], \tag{63}$$

which ultimately yields

$$A\gamma^s = \delta(1 - \gamma)(1 + A)\gamma^{\beta + r/c - r} \gamma^s. \tag{64}$$

Equating coefficients and applying this logic to each policy gives *Equation 7*:

$$A_n = \frac{\delta_n (1 - \gamma) \gamma^{\beta + r/c - r}}{1 - \delta_n (1 - \gamma) \gamma^{\beta + r/c - r}}.$$

The optimal policy (*Figure 1—figure supplement 11*) is given by (*Equation 8*) the sequence

$$\pi_n^* = \left( \frac{\delta_n (1 - \gamma)}{1 - \gamma^{s_n}} \right)^{1/r_n}.$$

We henceforth refer to $\delta_n$ as the aversion parameter. Large values of $\delta_n$ imply low aversion. It can be interpreted as the selected strategy in the simplest case where $\delta_n = \delta > 0$ and $s_n = r_n = 1$ since

$$\pi_n^* = \delta; \tag{65}$$

Rearranging **Equation 8** gives

$$\frac{1-\gamma_n^s}{1-\gamma} = \frac{\delta}{\pi_n^{*r_n}}. \tag{66}$$

## Solution uniqueness
## Proposition
The above value function is unique.

### Proof
We consider value functions $V(s)$ in the space of functions that are continuous in $\pi$ and bounded in $s$. We take $\|V\|_\infty \equiv \sup_s |V(s)|$. From the previous section, we have identified such a function $J$ so that

$$J(s_n) = \max_{0 \le \pi \le 1} \pi^r \left[ \gamma^{s_{n+1}} + (1 - \gamma^{s_{n+1}}) J(s_{n+1}) \right]. \tag{67}$$

Assume that $V(s)$ is another solution. For fixed $s_n$, let $\pi^*$ be such that

$$V(s_n) = \pi^{*r} \left[ \gamma^{s_{n+1}} + (1 - \gamma^{s_{n+1}}) V(s_{n+1}) \right]. \tag{68}$$

We can rewrite the following term:

$$\gamma^{s_{n+1}} = \gamma^{s_n - r + h_m \pi + \beta} = \gamma^{s_n - r + \beta} (\gamma^{h_m})^\pi \equiv \gamma^{k_s} \tilde{\gamma}^\pi, \tag{69}$$

where $\tilde{\gamma}, \gamma^{k_s} < 1$. Then

$$V - J = \pi^{*r} \left[ \gamma^{k_s} \tilde{\gamma}^{\pi^*} + (1 - \gamma^{k_s} \tilde{\gamma}^{\pi^*}) V(s_{n+1}) \right]$$
$$- \max_{0 \le \pi \le 1} \pi^r \left[ \gamma^{k_s} \tilde{\gamma}^\pi + (1 - \gamma^{k_s} \tilde{\gamma}^\pi) J(s_{n+1}) \right] \tag{70}$$

$$\le \pi^{*r} \left[ \gamma^{k_s} \tilde{\gamma}^{\pi^*} + (1 - \gamma^{k_s} \tilde{\gamma}^{\pi^*}) V(s_{n+1}) \right] - \pi^{*r} \left[ \gamma^{k_s} \tilde{\gamma}^{\pi^*} + (1 - \gamma^{k_s} \tilde{\gamma}^{\pi^*}) J(s_{n+1}) \right] \tag{71}$$

$$= \pi^{*r} (1 - \gamma^{k_s} \tilde{\gamma}^{\pi^*}) (V(s_{n+1}) - J(s_{n+1})) \tag{72}$$

$$\le \pi^{*r} (1 - \gamma^{k_s} \tilde{\gamma}^{\pi^*}) \left| V(s_{n+1}) - J(s_{n+1}) \right| \tag{73}$$

$$\le \pi^{*r} (1 - \gamma^{k_s} \tilde{\gamma}^{\pi^*}) \|V - J\|_\infty. \tag{74}$$

Note that

$$C(\pi) \equiv \pi^r (1 - \gamma^k \tilde{\gamma}^\pi) \le 1 - \gamma^k \tilde{\gamma}^\pi \tag{75}$$

is increasing in $\pi$ (since $\tilde{\gamma} < 1$) so that $C(\pi) \le 1 - \gamma^k \tilde{\gamma} \equiv K < 1$. Thus,

$$V - J \le K \|V - J\|_\infty. \tag{76}$$

By identical argument above, this time reversing the roles of $V$ and $J$ gives

$$J - V \le K \|V - J\|_\infty, \tag{77}$$

and so

$$\left| V(s_n) - J(s_n) \right| \le K \|V - J\|_\infty < \|V - J\|_\infty \quad \text{for all } s_n. \tag{78}$$

Therefore,

$$\|V - J\|_\infty = \sup_{s_n} |V(s_n) - J(s_n)| < \|V - J\|_\infty. \tag{79}$$

Thus,

$$\|V - J\|_\infty = 0. \tag{80}$$

□

## Mean optimal transitions

From *Equation 63*, the mean optimal transitions are

$$\mathbb{E}_n \left[ s_{n+1} \mid E_n \right] = s_n + (1/c - 1)r_n + \beta. \tag{81}$$

The mean increment, $\Delta s_n$, assuming the process is driven by $r_n \sim \text{Binomial}(s_n, q)$, becomes

$$\Delta s_n = (1/c - 1)qs_n + \beta. \tag{82}$$

We next consider two cases. In the first case, the basal antigen creation rate $\beta$ scales linearly with the number of currently recognized antigens, and in the second case we instead assume that it is fixed.

### $r_n$-linear basal antigen creation rate

This case considers $\beta = \alpha r_n$. Here, larger recognition in the current period results in larger exogenous penalty, and hence easier targeting, in the next period. Consequently, the number of detectable antigens in the future is directly influenced by both the tumor evasion strategy $\pi^*$ and the extent of that recognition resulting from immune targeting $r_n$. In this case (*Figure 5—figure supplement 1*), we have that

$$\mathbb{E} \left[ (1/c - 1 + \alpha)r_n \mid s_n \right] = (1/c - 1 + \alpha)qs_n, \tag{83}$$

so that the process satisfies the Martingale condition

$$\mathbb{E} \left[ s_{n+1} \mid s_n \right] = s_n \tag{84}$$

for critical alpha

$$\alpha_c = \frac{\log \gamma^{-1} - 1}{\log \gamma^{-1}}. \tag{85}$$

### Mutation accumulation rate

In the trivial case where, $\alpha = \alpha_c$, $s$ is constant and so mutation accumulation is predicted to be linear. Contributions by optimal evasion to the mutation rate are expected to exponentially decrease (resp. increase) over time if $\alpha < \alpha_c$ (resp. $\alpha > \alpha_c$).

In this case, dynamics and resultant mutation accumulation is determined by $\alpha$ relative to $\alpha_c$, and only those $\alpha$ close to the threshold generate behavior resembling linear mutation accumulation. Given this, the added penalty $\beta(r_n) = \alpha r_n$ due to the number of recognized antigens appears to be a less reasonable assumption based on empirical mutation rates (*Lawrence et al., 2013*; *Alexandrov et al., 2013*). We next consider the case for which the basal antigen creation rate is independent of $r$.

### $r_n$-independent basal antigen creation rate

In this case, $\Delta s_n$ from *Equation 83* becomes

$$\Delta s_n = (1/c - 1)qs_n + \beta. \tag{86}$$

The recognition dynamics of this case are more complex and partition into three regimes based on recognition relative to a critical threshold $q^* = 1 - 1/e$ (for which $c = 1$ and *Equation 87* $\Delta s_n = \beta$): effective immune recognition, critical recognition, and impaired recognition.

### Effective immune recognition

Here, $q > q^*$, giving $c > 1$. In this case, the Recognizer exerts a large recognition rate on the evading tumor. If $\beta \leq 0$, then the equilibrium, $s^*$ for which $\Delta s_n = 0$ is negative, and the $s_n$ is driven to 0. If $\beta$ is a positive, then there exists a stable, positive antigen state:

$$s^* = \frac{\beta}{q(1-1/c)} \tag{87}$$

Trajectories assuming a variety of initial conditions are given with $s^* = 10$ in *Figure 5—figure supplement 2A*.

## Impaired immune recognition

In contrast with effective recognition $q < q^*$, $c < 1$, and in this case, the equilibrium points are unstable. Moreover, If $\beta \geq 0$, then by a similar reasoning as above, $s^* \leq 0$ so that $s_n$ is driven to become very large. Alternatively, if $\beta < 0$ then the equilibrium state is

$$s^* = \frac{\beta}{q(1/c-1)} \tag{88}$$

so that collectively the equilibrium value is given by *Equation 10*.

## Critical immune recognition

At criticality $q = q^*$, $c = 1$, and *Equation 83* simplifies to

$$\Delta s_n = \beta. \tag{89}$$

In this special case, all randomness imparted to the process by $r_n$ is eliminated by a critical offset in the number of recognized antigens and the net addition of new antigens so that the long-term behavior of the process is completely determined by $\beta$. Predictably, $\beta > 0$ (resp. $\beta < 0$) results in net expansion (resp. depletion) of antigens over time, and $\beta = 0$ is stationary. The sign of $\beta$ may change as a function of the tumor IME. For example, immune exclusion and the resulting attenuated inflammation may both decrease $q$ and $\beta$ as well as genetic aberrations involving mismatch repair (MMR) deficiency and microsatellite instability. Other alterations, such as modulated MHC expression, or MHC loss of heterozygosity (LOH), may affect $q$ in isolation *Rosenthal et al., 2019*.

## Mutation accumulation rate

Critical and impaired immune recognition dynamics follow a similar behavior to that detailed in Section Mean optimal transitions. The effective recognition case bears a resemblance to the approximate dynamics of the informed Evader in Section Mean transition behavior. Here, by a similar argument in Section Mutation accumulation rate and tumor antigen availability once equilibrium is achieved, we have that

$$\nu(n) \equiv \frac{\Delta\lambda(n)}{\Delta n} \propto r_n + \beta + f_n. \tag{90}$$

Studying the process at $s_0 = s^*$ given by *Equation 88*, and $f_n^* = r_n/c$, we have that

$$
\begin{aligned}
\mathbb{E}_n\left[\nu(n) \mid E_n\right] &\propto \mathbb{E}_n\left[r_n + r_n/c + \beta \mid E_n\right] \\
&= (1+1/c)\mathbb{E}_n\left[r_n \mid E_n\right] + \beta \\
&= (1+1/c)qs^* + \beta \\
&= \beta\left[\frac{(1+1/c)}{(1-1/c)} + 1\right] \\
&= \left(\frac{2c}{c-1}\right)\beta.
\end{aligned}
\tag{91}
$$

This implies *Equation 11*:

$$\lambda(n) \propto 2\beta cn/(c-1).$$

Therefore, linear mutation accumulation as a function of time ensues for an effective Recognizer as in the passive Evader case (*Equation 46*), this time as a function not only of the basal antigen creation rate $\beta > 0$ but also of $q$ through $c$. We recall that under effective recognition, $q^* < q < 1$ (equivalently $1 < c < \infty$), which ultimately gives via *Equation 11*

$$2\beta n < \mu(n). \tag{92}$$

## Dynamics summary

The assumption that the basal antigen production depends on recognition $\beta = \alpha r_n$ results in exponential growth or decay in the number of recognizable antigens (and therefore mutation rate), and it was only for a very narrow parameter value $\alpha \sim \alpha_c$ for which linear mutation accumulation could occur. It is for this reason that the $r_n$-linear constitutive assumption is less realistic.

For basal antigen rates $\beta$ that are $r_n$-independent, mutations are predicted to accumulate linearly under effective immune recognition, in a similar manner to that observed in the passive Evader case. In contrast with that case, however, an active Evader executes an optimal strategy to maximize the overall escape probability. This predicts that one effect of a dynamic evasion that optimally maximizes escape probability is a concomitant increase in the mutation accumulation rate relative to the passive case via a correction term $c/(c-1)$. This enhancement becomes indistinguishable when recognition is very aggressive ($q \to 1$) and becomes large when $q$ approaches the critical detection rate.

Interestingly, the active evasion strategy predicts that mutation accumulation rates vary as a function of recognition pressure, in contrast with the passive evasion model. Additionally, disease progression may affect immune recognition (changes in $q$) and tumor evasion penalty (changes in $\beta$). While the number of recognizable TAAs for the passive case continues evolve according to the mean-reverting process, there is a dramatic discontinuity in active systems whereby recognition rates below a critical threshold may result in unstable behavior prior to escape (*Figure 5—figure supplement 2*).

## Optimal evasion strategy

From *Equations 6–8*, we have

$$J(s_n, r_n) = \frac{A_n e^{-cs_n}}{1 - e^{-cs_n}}, \tag{93}$$

$$A_n = \frac{\delta_n q e^{-(1-c)r_n - c\beta}}{1 - \delta_n q e^{-(1-c)r_n - c\beta}}, \tag{94}$$

and

$$\pi_n^* = \left( \frac{\delta_n q}{1 - (1-q)^{s_n}} \right)^{1/r_n}. \tag{95}$$

Thus,

$$J(s_0, r_0) = \frac{\delta_0 q e^{-(1-c)r_0 - c\beta}}{1 - \delta_0 q e^{-(1-c)r_0 - c\beta}} \cdot \frac{e^{-cs_0}}{1 - e^{-cs_0}}, \tag{96}$$

We note that for $s_n = s_{n-1} + (1/c - 1)r_{n-1} + \beta$, therefore

$$
\begin{aligned}
e^{-cs_n} &= \gamma^{s_{n-1} + (1-c)r_{n-1}/c + \beta} \\
&= \gamma^{s_{n-2} + (1-c)(r_{n-1} + r_{n-2})/c + 2\beta} \\
&= \dots = \gamma^{s_0 + n\beta + \mathbb{C}_\gamma R_{n-1}},
\end{aligned}
\tag{97}
$$

where

$$\mathbb{C}_\gamma \equiv \frac{1 - \ln \gamma^{-1}}{\ln \gamma^{-1}} \tag{98}$$

and

$$R_n \equiv \sum_{j=1}^{n} r_j. \tag{99}$$

By iteratively applying *Equation 98*, we ultimately obtain the value function in terms of the history of the environmental landscape, $\{r_n\}_n$

$$J(s_n, r_n) = \frac{\delta_n q (1-q)^{\mathbb{C}_\gamma r_n - \beta}}{1 - \delta_n q (1-q)^{\mathbb{C}_\gamma r_n - \beta}} \cdot \frac{(1-q)^{s_0 - n\beta + \mathbb{C}_\gamma R_{n-1}}}{1 - (1-q)^{s_0 - n\beta + \mathbb{C}_\gamma R_{n-1}}}. \tag{100}$$

We remark that this simplifies for constant $\delta_n = \delta$, which we will typically take as 1.

## Critical recognition

At the critical value of recognition $q^* = 1 - 1/e$ ($c = 1$), the dynamics become deterministic. Here, the value of the present state depends only on the initial number of detectable antigens and number of periods that have elapsed and is independent of the history of recognized antigens $\{r_n\}_n$.

$$J(s_n, r_n) = \frac{\delta_n q(1-q)^\beta}{1 - \delta_n(1-q)^\beta} \cdot \frac{(1-q)^{s_0 - n\beta}}{1 - (1-q)^{s_0 - n\beta}}. \tag{101}$$

At criticality, the value of the present state depends only on the initial number of detectable antigens and number of periods that have elapsed, and not on the number of recognized antigens.

## Non-critical recognition

We recall that the value function carries meaning as the maximal attainable expected future value. Under effective recognition ($c = 1 \Rightarrow \gamma^{\mathbb{C}r}$ is increasing in $r$), so that the value function (*Equation 101*) has an exponent that increases.

We are motivated to consider either mild or aggressive recognition of Section 5.2.4. We will assume that there is minimal aversion so that $\delta_n = 1$.

## Predicted dynamical behavior

From Section Mean optimal transitions, the dynamical behavior of the number of recognizable TAAs, or immunogenicity, of an active Evader is determined by $\beta$ and $q$. Disease progression may ultimately affect immune recognition (reducing $q$) and/or tumor basal tumor antigen creation (reducing $\beta$). $\beta$ is expected to vary widely across tumor types. Within a given tumor subtype, the extent of environmental hostility is expected to require additional tumor adaptation that may manifest as additional TAA targets. Therefore, larger (resp. smaller) evasion penalties $\beta$ correspond with *anti-tumor* (resp. *pro-tumor*) IME. Similarly, larger (resp. smaller) $q$ corresponds to *infiltrated* (resp. *excluded*) environments, and from this we model four possible states: anti-tumor-infiltrated, anti-tumor-excluded, pro-tumor-infiltrated, and pro-tumor-excluded. The model predicts that infiltrated ($q > q^*$) environments lead to an absorbing equilibrium state in the intervening period prior to escape, while exclusion ($q < q^*$) result in unstable equilibria. Interestingly, the sign of the equilibrium, and hence the behavior, depends on $\beta$, and leads to dramatically diverse behavior in the antigenicity of a dominant tumor clone as it progresses via immune recognition. This case is meaningful as long as the intertemporal penalty assuming the optimal strategy occurs, $\beta + f_n^*$, remains non-negative whenever there is at least one recognition event. This is equivalent to the condition that $f_n^* + \beta \geq 1/\ln \gamma^{-1} + \beta > 0$, which is assumed in all examples that follow. These results are summarized in *Figure 5* and organized below. The corresponding immunogenicity and cumulative mutations following escape are given by *Figure 4*, with the timing of escape and example trajectories given by *Figure 5—figure supplement 3*.

1. Anti-tumor-infiltrated ($q > q^*$, $\beta > 0$): This stable steady state is positive, so that the process is mean-reverting, and generates immunogenically warm' tumors.
2. Anti-tumor-excluded ($q < q^*$, $\beta > 0$): Here, recognition is low, while the arrival of new TAAs is large. This unstable steady state is negative, so that all trajectories tend to increase their immunogenicity over time, resulting in 'hot' tumors.
3. Pro-tumor-infiltrated ($q > q^*$, $\beta < 0$): In this case, recognition is large while the arrival of new TAAs is low. This stable steady state is negative, so that all trajectories tend to reduce their immunogenicity to zero over time, yielding 'cold' tumors.
4. Pro-tumor-excluded ($q < q^*$, $\beta < 0$): Lastly, if both recognition and new TAA arrival rates are low, then there is a positive unstable state, above which trajectories accumulate additional TAAs over time, becoming 'hot,' and below which the populations are predicted to reduce the number of recognizable TAAs over time, becoming 'cold.'

These predicted dynamics parallel the observation that tumors under active immunosurveillance via effective recognition undergo significant immunoediting. Our results predict that the resulting tumor becomes 'warm' or 'cold' depending on the extent of new TAA arrival during active evasion. On the one hand, impaired recognition leads to diverse behavior dependent on the rate at which new TAAs are acquired during active evasion. If this acquisition rate is large, then the tumor accumulates

TAAs over time to become 'hot.' On the other hand, tumors subject to reduced selection pressures may evolve as immune-hot or immune-cold tumors, consistent with previous observations (*Lakatos et al., 2020*). Moreover, the effect of reducing immune recognition leads to an accumulation of TAAs over time, consistent with experimental observations in lung cancer wherein patients with HLA loss of heterozygosity harbored larger mutational burdens, an indirect measure of TAA number of our model (*McGranahan and Swanton, 2017*). Our predictions suggest that immunogenicity ultimately depends on the number of detectable TAAs at the time of impaired immune recognition, suggesting that TAA-depleted tumors share in common the tendency for their evasion strategies to incur less antigenic penalties. Our results would predict the utility of altering the tumor microenvironment to increase the immunogenicity of immune-cold tumors by making evasion more costly in a manner reminiscent of mutational meltdown (*Gabriel et al., 1993*). We remark that these dynamics are worth considering in the case of adoptive T cell-based immunotherapies, which have a large potential for exerting substantial co-evolutionary pressure on a developing malignancy (*George and Levine, 2021*).

## Survival benefit of active evasion

From the above analysis, immunogenicity dynamics of an active Evader are closest to those of a mean-reverting passive Evader under the pro-tumor-infiltrated case. Given this, we study the dynamics under active and passive evasion as well as the distribution of escape times and probability of escape (*Figure 2*). For a reasonable comparison, we fix $q$ and $s^*$ for each case, and the passive evasion rate $p$ is chosen to match the stationary mean optimal evasion rate $\pi^*$. Our simulations result in escape occurring 1.6 times more frequently under active evasion. Moreover, active evasion exhibits a broader distribution of elimination and escape times (Mean Passive Escape = 6.0, Var Passive Escape = 25.0, Mean Passive Elimination = 6.1, Var Passive Elimination = 30.1; Mean Active Escape = 7.2, Var Active Escape = 35.8, Mean Active Elimination = 6.7, Var Active Elimination = 38.0). Our results demonstrate that active evasion allows an Evader to adapt to the observed recognition and, despite continual penalty, allows an Evader to 'out-wait' a Recognizer in order to undergo escape.

## Exogenous recognition

One powerful advantage of this approach is that the theoretical predictions are not limited by the underlying distribution of $r_n$ driving the process. In fact, the optimal policies and value function can handle any temporally varying recognition landscape, $\{r_n\}_n$, so long as $0 \leq r_n \leq s_n$. We consider the effects of step, cyclical, increasing, and decreasing recognition landscapes on the relative evasion probability for populations adopting either a passive or active strategy (*Figure 3*).

In addition to arbitrary recognition landscapes, our dynamic programming approach may be applied to understand the effects of immunotherapeutic intervention, whereby immune escape can be modeled as a range of possible behavior on the spectrum of passive evasion to the most aggressive (active) evasion. For example, the active evasion dynamics assuming an anti-tumor-infiltrated case are similar to those of passive evasion. In both cases, the process escapes with immunogenicity values that fluctuate around a stationary $s^*$. We can recover the recover the relationship between $s^*$ and mutation rate $\nu(n) = \Delta\lambda/\Delta n$ via *Equations 32 and 46* for the passive case and *Equation 88*, *Equation 11* for the active case. In both cases, the result is similar:

$$s^* = \nu/2q. \tag{102}$$

demonstrating that immunogenicity, and thus the success likelihood of immunotherapeutic intervention, varies directly with mutation rate and inversely with recognition rate. This theory predicts that escape to a cold tumor is more likely when $s^*$ is close to 0 and is akin to complete evasion as modeled in *George and Levine, 2018*, contrasting with temporary evasion that may be recognized subsequently *George and Levine, 2020*. All else equal, higher mutational rates can lead to higher predicted efficacy via higher $s^*$, but this is not the only way as concomitantly high rates of recognition can drive $s^*$ down, thereby reducing predicted efficacy. In *Equation 103*, it is clear that a better immunotherapy prognosis occurs when the mutational rate is higher and the recognition rate is also

low since $s^*$ is predicted large in this case. *Figure 5—figure supplement 4* summarizes the behavior of an adaptive Evader subject to a temporally varying recognition pressure.

## Acknowledgements

JTG thanks Kerry E Back, Philip A Ernst, Thomas J George, and Richard A Tapia for their helpful discussions on stochastic dynamic programming and optimization. JTG was supported by the Cancer Prevention Research Institute of Texas (RR210080). JTG is a CPRIT Scholar in Cancer Research. HL is supported by the National Science Foundation (NSF) grant NSF PHY-2019745.

## Additional information

### Funding

| Funder | Grant reference number | Author |
|---|---|---|
| Cancer Prevention Research Institute of Texas | RR210080 | Jason T George |
| National Science Foundation | PHY-2019745 | Herbert Levine |

The funders had no role in study design, data collection and interpretation, or the decision to submit the work for publication.

### Author contributions

Jason T George, Conceptualization, Formal analysis, Supervision, Funding acquisition, Investigation, Methodology, Writing – original draft, Project administration, Writing – review and editing; Herbert Levine, Supervision, Funding acquisition, Investigation, Writing – original draft, Project administration, Writing – review and editing

### Author ORCIDs

Jason T George (iD) http://orcid.org/0000-0002-8248-2888

### Decision letter and Author response

Decision letter https://doi.org/10.7554/eLife.82786.sa1
Author response https://doi.org/10.7554/eLife.82786.sa2

## Additional files

### Supplementary files

• MDAR checklist

### Data availability

All data generated or analyzed in this study are included in the supplementary data files. Source code is publicly available as a git repository (*George, 2022*).

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
