## [Editor Report]

This study presents a valuable mathematical model for the adaptive dynamics of cancer evolution in response to immune recognition. The mathematical analysis is rigorous and convincing, and overall the framework presented could be used in the future as a solid base for analytically tracking tumor evasion strategies. The work will be of interest to evolutionary cancer biologists and potentially may also have implications for the design of clinical interventions.

---

## [Decision Letter]

**Decision letter after peer review:**

Thank you for submitting your article "Optimal Cancer Evasion in a Dynamic Immune Microenvironment Generates Diverse Post-Escape Tumor Antigenicity Profiles" for consideration by *eLife*. Your article has been reviewed by 3 peer reviewers, one of whom is a member of our Board of Reviewing Editors, and the evaluation has been overseen by Aleksandra Walczak as the Senior Editor. The reviewers have opted to remain anonymous.

Essential revisions:

As you can see below, the scope and mathematical effort in your manuscript were greatly appreciated by all reviewers. The model is mathematically rigorous and addresses an important and timely problem. There were some doubts about the applicability of your model to cancer evolution in real situations, but as a conceptual paper, it can contribute much to the advancement of the field. Even then, it needs some more work to be truly beneficial for the community.

1) Clarity of the paper should be improved. This includes a better discussion of the underlying assumptions, less technical terms and clearer ones, and a tighter exploration of the important parameters of the model, possibly with a phase diagram or similar graphical aid.

2) The process of Clonal cancer evolution should be better discussed in relation to the model, which describes the dynamics of mutations but does not follow clones.

3) The claim that the cancer cells are sensing the immune system is a bold and intriguing one, and as such need to be better supported, or otherwise, it is hard to justify.

*Reviewer #1 (Recommendations for the authors):*

The actual evolutionary process in the tumor happens on the level of cells and clones. Tumor clones proliferate and compete, each with its own fitness under the changing selection pressure from the immune system. However, the model here does not address tumor cells or clones, but rather the presented antigens as independent particles. While this is not an unreasonable choice, it does require better discussion and justification.

A major ingredient of the model is the penalty for immune evasion – in which while evading immune recognition of certain TAAs, a tumor will develop new ones as potential new targets for the immune system. This is presented in an opaque way, with several assumptions and specific mathematical forms. There should be some tradeoff between immune escape and loss of function, that would lead to such a penalty, and it should be made more concrete and transparent.

The idea that cancer can sense and respond to the immune system and the tumor microenvironment is exciting and intriguing but therefore requires stronger evidence. It would be valuable to make more connections to known observations to support this type of model. This is especially true since some of the model "predictions", like the correlation between lower immune surveillance and tumor mutational burden, are known from simpler principles.

And more generally, language and notation should be improved throughout the manuscript, making it easier to follow. Some jargon should be discouraged or explained, like "date-n" or "mean evolution dynamics" (section 3.1). A phase-space-like diagram of the different parameters' regimens of the model (maybe β and q?) would also be extremely useful.

The definition of eta is confusing. The manuscript states: 'eta may be interpreted as the probability of the complement of the following event: "recognition occurs without matched evasion for a single antigen". In other words, eta is the probability of a tie at a single antigen position.' But the complement of that event also includes the probability of no recognition, on top of a tie, unless I misunderstood what tie means. Regardless, tie is a confusing term here and should at least be explained better.

*Reviewer #2 (Recommendations for the authors):*

*Reviewer #3 (Recommendations for the authors):*

Developing such conceptual models is important and has the potential to inspire the wider field. However, I fear that some of that full potential might not be reached without additional work rewriting the text for greater clarity and precision. The following are some suggestions that might be helpful in that regard.

To start with, I had a hard time following some of the terminology. There are some instances where different terms are used to refer to the same concept. For instance, in Figure 1C the terminology changes between the legend (evasion rate) and plot (optimal downregulation attempt probability). Furthermore, the terminology seems overloaded: π is referred to as an evasion probability, but maybe one would want to reserve this term for the complete evasion of immune recognition by cancer. π might be more simply referred to as a rate of antigen loss, or similar. I was also wondering whether it would make sense to directly include β in Eq.1 and separate it from the penalty term. I understood penalty to refer to the increase in antigen creation rate when π is higher. From the equation, β instead could be more aptly named a basal rate of antigen creation. Lastly, notations should be uniformized. Specifically, I noted that in the methods the rate at which TAAs are lost is denoted by p instead of pi, if I understand correctly, which can cause confusion for the reader.

---

## [Author Response]

Essential revisions:As you can see below, the scope and mathematical effort in your manuscript were greatly appreciated by all reviewers. The model is mathematically rigorous and addresses an important and timely problem. There were some doubts about the applicability of your model to cancer evolution in real situations, but as a conceptual paper, it can contribute much to the advancement of the field. Even then, it needs some more work to be truly beneficial for the community.1) Clarity of the paper should be improved. This includes a better discussion of the underlying assumptions, less technical terms and clearer ones, and a tighter exploration of the important parameters of the model, possibly with a phase diagram or similar graphical aid.

We share in the reviewer’s enthusiasm for our model’s mathematical rigor and conceptualization, and we are grateful for the helpful suggestions on improving model clarity.

Following the reviewers’ suggestions, we have made several significant changes to the manuscript to clarify the underlying assumptions. First, we have changed the language describing the evasion probability and the rate of Tumor-Associated Antigen (TAA) loss (π_n_). We have also re-structured the presentation of the evasion penalty to separate out the decision-dependent (π_n_) term, now given by *f_n_*, from the decision-independent contribution given by the β which is more aptly referred to as the basal antigen creation rate throughout the manuscript. The first use of fixed π*_n_* equal to *p* has been clarified in the Methods section.

Two of the reviewers had suggested that there is a degree of confusion in discussing the *tie probabilities* in this process. In an effort to clarify our approach, we now replace that language with the idea of *tumor-immune equilibrium* in order to keep the discussion grounded in Schreiber’s widely accepted conceptual framework of the ‘Three E’s (Elimination, Escape, Equilibrium) of Immunoediting’. The language has now been changed in the Results and Methods sections and described explicitly in the Introduction:

“Immunosurveillance via distinct T cell clones imposes an adaptive, stochastic recognition environment on a developing cancer population (Desponds et al. *PNAS* 2016) that can result either in cancer elimination, escape, or equilibrium (Dunn et al. *Annu Rev Immunol* 2004). Equilibrium results in cancer co-existence with the immune system over large time scales (Turajlic et al. *Cell* 2018) thereby motivating the need for a more complete understanding of the interplay between immune recognition and cancer evolution for effective therapeutic design.”

We appreciate the editor’s suggestion of a phase diagram to graphically depict the important parameter regions in question and have now added this graphical aid to Figure 5 to clarify the relevant regimes corresponding to those cases.

2) The process of Clonal cancer evolution should be better discussed in relation to the model, which describes the dynamics of mutations but does not follow clones.

It has been correctly noted that our model does not attempt to deal with possible subclonal structure of the tumor. This possibility introduces additional complications into the model formulation and we wanted to first establish the basic idea of “adaptive evasion” in the simplest context. In an attempt to illustrate how reasonable future generalizations of our model could include nontrivial sub-clonal heterogeneity in tumor antigens tracked through time, we now describe how one would go about enhancing the existing model to address this. The following has been added to the Methods and Discussion sections:

Addition to Methods: “The above describes a clonal population harboring a core minimal set of TAAs for which recognition and down-regulation ultimately determine cancer escape, elimination, or equilibrium. Our model can however be adapted to study the more general scenario involving a clonal hierarchy of heterogeneous cancer cells. We illustrate this by considering a population of cells with a set C of c = | C | core clonal TAAs, together with distinct groups of cells with subclonal collections of TAAs S_1_ and S_2_ (having size s_1_ = |S_1_| and s_2_ = |S_2_|, respectively). The relevant populations therefore have antigen sets given by P_1_ = CUS_1_ and P_2_ = CUS_2_. The basic event considered in the foundational model, [r_n_ > 0], must now be replaced by the event that recognition occurs in both P_1_ and P_2_; in the absence of recognition of both subclones, the cancer escapes. Recognition happens either if there is a recognition event r in C or if there are simultaneous recognition events r_1_ in S_1_ and r_2_ in S_2_. Assuming that TAA recognition occurs independently as before with probability q, the total probability of relevant recognition, originally (1−γ^sn^), is now given by (1−γ^c^) +γ^c^(1−γ^r1^)(1− γ^r2^). The first term characterizes the coupling of the fate of both subclones should a common TAA be recognized, while the latter term represents the parallel recognition process required to control each subclone separately via subclonal TAA recognition. Lastly, assuming that recognition proceeds either by a shared TAA in C or instead by subclonal TAAs in both S_1_ and S_2_, then the probability of elimination and progression proceed identically as before. In the remainder of the discussion, we will, for baseline understanding, only track a core set of clonal antigens on the fittest clone.”

*Addition to Discussion: “*In this foundational model, we demonstrated the dynamics of immune recognition of an adaptive population of cancer cells expressing a purely clonal pattern of antigens. Our model implicitly equates antigen loss and the progression of a subpopulation currently adapted to evade immune targeting – either by direct pruning of the fittest subclone or by stochastic emergence and subsequent growth of a new one lacking the targeted antigens – as equivalent. Here, we tracked the fittest clone represented by a core set clonal antigens. We remark that heterogeneous populations each having a distinct sub-clonal signature can also be tracked, but the corresponding antigen-driven selection and fitness cost to each clone would be coupled through shared antigens (see Methods). Finally, we note that this extended approach implicitly assumes that antigen detection rates over a given period are subclone size-independent, given that antigens are tracked over a period where each of the clones with comparable fitness would be detectable by the immune system during their growth trajectory en route to attempted escape.”

3) The claim that the cancer cells are sensing the immune system is a bold and intriguing one, and as such need to be better supported, or otherwise, it is hard to justify.

We are grateful to the editor and reviewers for requesting that this point be made more explicitly. The rheostat for stress that was previously discussed in the initial draft was specific for stress responses that lead to increased mutagenesis in yeast and hypoxia-driven responses in cancer. While this is indicative of a cell’s capability of an adaptive response, it did not specifically relate this possibility to the actions of the immune system. What we have now included is a direct connection between T-cell activity and this response, which is known to occur both indirectly, in the setting of immune-mediated cytokine release following T cell recognition, and as a result of T cell targeting directly. In an attempt to present concrete connections to empirical observations in support of this phenomenon, we have added the following paragraph to the Discussion section:

“Our analysis centered on the ability of cancer populations to adaptively respond to a measured immune state, and we have primarily focused on studying subsequent mutations resulting in the disruption of existing (targeted) tumor-associated antigenic targets and on the generation of new ones. It is important to note that independent empirical observations support the ability of cancer cells to sense their immune microenvironment (IME), and perhaps even the level of CD8^+^ killing that occurs therein. At the signaling level, IL-6 secreted by CTLs, macrophages, and dendritic cells in response to immune recognition has been shown to directly activate ataxia-telangiectasia mutated (ATM), a factor implicated in response to DNA damage, and this has been associated with increased metastasis and multi-drug resistance in lung cancer (Jiang et al. *Oncotarget* 2015; Yan et al. *Cancer Science* 2014). IFN-γ released by activated CD8^+^ tumor-infiltrating lymphocytes activates the cell-intrinsic STING pathway in response to DNA damage in cancer, implicating an altered TME from activated CD8^+^ T-cells that is measurable by the cancer (Xiong et al. *Oncoimmunology* 2022). Lastly, at the level of individual TCR interactions with recognized tumor cells, Granzyme B release has been directly linked to DNA damage and associated CHK2 and p53 stress-responses, and studies have demonstrated hSMG-1 stress-activated proteins upregulated in cancer cells following granzyme B treatment (Meslin et al. *J Mol Med* 2011). Moreover, granzyme release in the microenvironment serves a signaling molecule promoting a pro-inflammatory response from other immune cells (Cullen et al. *Cell Death Differentiation* 2010). The relatively acute response and short half-lives of downstream effectors (minutes for p53 and hours for CHK1, for example), provide a tunable response based on the current level of immune targeting through stress-induced mutagenesis (Bindra et al. *Cancer Met Rev* 2007; Rosenberg *Nat Rev Genetics* 2001; Rosenberg, Queitsch. *Science* 2014) that in our analysis directly influences tumor-associated antigen availability.”

Reviewer #1 (Recommendations for the authors):The actual evolutionary process in the tumor happens on the level of cells and clones. Tumor clones proliferate and compete, each with its own fitness under the changing selection pressure from the immune system. However, the model here does not address tumor cells or clones, but rather the presented antigens as independent particles. While this is not an unreasonable choice, it does require better discussion and justification.

We thank the reviewer for their helpful comments. We remark that our model implicitly equates antigen loss and the progression of a subpopulation currently adapted to evade immune targeting – either by direct pruning of the fittest subclone or by stochastic emergence and subsequent growth of a new one lacking the targeted antigens – as equivalent. The foundational analysis was conducted on a clonal antigenic structure assuming a minimal collection of TAAs that required targeting for a change in equilibrium to either escape or elimination. Because we for foundational understanding studied the case where a single clonal signature was tracked in time, we under-explained the implementation of such a model in more complicated cases.

The next most complicated scenario involves a heterogeneous population of cancer cells with disjoint neoantigen profiles. In this case, a parallel process can be studied wherein the effects of recognition in one environment are decoupled from the other (relevant to, for example, spatially distinct subpopulations). This description however misses the case where such disparate populations evolve to express shared antigens, or in the case where there are both clonal and subclonal antigen targets. Here, our model can still be applied in parallel to study distinct clones but requires additional structure. Namely, in this case we would need to incorporate non-trivial coupling between the possible recognition/selection against certain antigens shared across clones. For example, control of a population with clonal antigens {*a, b*} but having unique subclones having either antigens {*w,x*} or {*y*,*z*} could be considered by studying the process in parallel, and control in the next periods would require recognition/selection against either (1) at least one of {w,x} *and at least one of* {*y,z*}, or (2) at least one of {*a*,*b*}. In this more general framework, the arrival of new subclones with distinct features from the parent clone in question could also be incorporated and studied across time periods. This strategy of subdividing more complicated evolutionary structures has now been further elaborated on in the Methods section, and we have expounded these points in the discussion (see additions given under Editor Comment 2).

A major ingredient of the model is the penalty for immune evasion – in which while evading immune recognition of certain TAAs, a tumor will develop new ones as potential new targets for the immune system. This is presented in an opaque way, with several assumptions and specific mathematical forms. There should be some tradeoff between immune escape and loss of function, that would lead to such a penalty, and it should be made more concrete and transparent.

Indeed, the clarity of this balance between TAAs both lost and generated was discussed in an excessively complicated way. In order to add more clarity, we have isolated the basal arrival of TAAs from the penalty term. In doing so, the assumptions on the functional form for penalty is simplified to be directly proportional to the rate of TAA loss and distinct from any r_n_,π_n_-independent TAA additions fully characterized by the β term. We have also clarified these points early in the model development section at the point where β and penalty terms *f*_n_ are described: XXX

The idea that cancer can sense and respond to the immune system and the tumor microenvironment is exciting and intriguing but therefore requires stronger evidence. It would be valuable to make more connections to known observations to support this type of model. This is especially true since some of the model "predictions", like the correlation between lower immune surveillance and tumor mutational burden, are known from simpler principles.

We thank the reviewer for this helpful suggestion since our original presentation omitted all but one sentence on the precise molecular details of how this may occur. The rheostat on stress that was previously discussed was specific for stress responses that lead to increased mutagenesis in yeast and hypoxia-driven responses in cancer. What should have been included was a direct connection between T-cell activity and this response, which is known to occur both indirectly, in the setting of immune-mediated cytokine release following T cell recognition, and as a result of T cell targeting directly. In an attempt to make concrete connections to empirical observations in support of this phenomenon, we have added these points in the Discussion section (see additions given under Editor Comment 3).

And more generally, language and notation should be improved throughout the manuscript, making it easier to follow. Some jargon should be discouraged or explained, like "date-n" or "mean evolution dynamics" (section 3.1). A phase-space-like diagram of the different parameters' regimens of the model (maybe β and q?) would also be extremely useful.

We thank the reviewer for suggesting helpful additions to aid in presentation clarify. Specifically, we have removed excessive jargon (‘date-*n*’) and have expanded upon what we mean by mean evolution dynamics. The thought of adding a phase-space diagram for a visual map of the model’s key regimes is a great suggestion, and we have now added this to Figure 5.

The definition of eta is confusing. The manuscript states: 'eta may be interpreted as the probability of the complement of the following event: "recognition occurs without matched evasion for a single antigen". In other words, eta is the probability of a tie at a single antigen position.' But the complement of that event also includes the probability of no recognition, on top of a tie, unless I misunderstood what tie means. Regardless, tie is a confusing term here and should at least be explained better.

*ղ* = 1−*q*(1−*p*) is by definition the probability complement of *q*(1−*p*), which is the probability for a single recognition event without antigen loss (at that one position). The reviewer is correct that this event can occur either via no antigen recognition (so a ‘tie’ occurs at that position by default), or by recognition at that position being matched by antigen loss.

To make this clearer and to remove the ‘tie’ interpretation out of the description, we now explain that balanced recognition and evasion is referred to as ‘equilibrium’, which needs to occur for any and all antigens that are recognized for the process to avoid escape or elimination, but that we can also characterize the probability that a single antigen is in equilibrium. In this way, *ղ* represents the probability that equilibrium exists in one antigen position provided that there is at least one available antigen for immune targeting.

Reviewer #3 (Recommendations for the authors):Developing such conceptual models is important and has the potential to inspire the wider field. However, I fear that some of that full potential might not be reached without additional work rewriting the text for greater clarity and precision. The following are some suggestions that might be helpful in that regard.To start with, I had a hard time following some of the terminology. There are some instances where different terms are used to refer to the same concept. For instance, in Figure 1C the terminology changes between the legend (evasion rate) and plot (optimal downregulation attempt probability). Furthermore, the terminology seems overloaded: π is referred to as an evasion probability, but maybe one would want to reserve this term for the complete evasion of immune recognition by cancer. π might be more simply referred to as a rate of antigen loss, or similar. I was also wondering whether it would make sense to directly include β in Eq.1 and separate it from the penalty term. I understood penalty to refer to the increase in antigen creation rate when π is higher. From the equation, β instead could be more aptly named a basal rate of antigen creation. Lastly, notations should be uniformized. Specifically, I noted that in the methods the rate at which TAAs are lost is denoted by p instead of pi, if I understand correctly, which can cause confusion for the reader.

We thank the reviewer for their thorough review and thoughtful suggestions. We have addressed some the issues above, which we believe enhances the clarity of the new draft. In taking the reviewer’s suggestion, we now describe π as the rate of antigen loss, reserving ‘evasion probability’ for discussions involving complete immune evasion.

We very much like the suggestion of isolating/emphasizing the β basal rate of antigen creation from the penalty (in the strict sense) incurred for choosing larger π_n_. This is a clearer way to present the Results section that already assumed β to be *r*_n_,π*_n_*-independent. Toward this end, we have added this to Eq. 1 and discuss it thereafter, and we have made the additional updates to the other Results sections where β and *f_n_* are mentioned.

Our reason for describing the TAA loss rate with *p* and π*_n_* separately is to emphasize when it is simply a passive feature (π_*n*_ = *p* fixed) versus when it is actively chosen. This was distinguished in the Model Development section but was previously confusing given the initial use of *p* followed by π*_n_*. For clarity, we have now elaborated at the first mention of the antigen loss rate: “The rate of antigen loss π*_n_* may in general vary as a function of time and environmental features (considered in Sec. 5.2). In this section, we assume it is passively fixed and denote this rate as *p*.